# Near-Optimal $k$-Clustering in the Sliding Window Model

**David P. Woodruff**
CMU
dwoodruf@cs.cmu.edu

**Peilin Zhong**
Google Research
peilinz@google.com

**Samson Zhou**
Texas A&M University
samsonzhou@gmail.com

## Abstract

Clustering is an important technique for identifying structural information in large-scale data analysis, where the underlying dataset may be too large to store. In many applications, recent data can provide more accurate information and thus older data past a certain time is expired. The sliding window model captures these desired properties and thus there has been substantial interest in clustering in the sliding window model.

In this paper, we give the first algorithm that achieves near-optimal $(1 + \varepsilon)$-approximation to $(k, z)$-clustering in the sliding window model, where $z$ is the exponent of the distance function in the cost. Our algorithm uses $\frac{k}{\min(\varepsilon^4, \varepsilon^{2+z})}$ polylog $\frac{n\Delta}{\varepsilon}$ words of space when the points are from $[\Delta]^d$, thus significantly improving on works by Braverman et. al. (SODA 2016), Borassi et. al. (NeurIPS 2021), and Epasto et. al. (SODA 2022).

Along the way, we develop a data structure for clustering called an online coreset, which outputs a coreset not only for the end of a stream, but also for all prefixes of the stream. Our online coreset samples $\frac{k}{\min(\varepsilon^4, \varepsilon^{2+z})}$ polylog $\frac{n\Delta}{\varepsilon}$ points from the stream. We then show that any online coreset requires $\Omega\left(\frac{k}{\varepsilon^2} \log n\right)$ samples, which shows a separation from the problem of constructing an offline coreset, i.e., constructing online coresets is strictly harder. Our results also extend to general metrics on $[\Delta]^d$ and are near-optimal in light of a $\Omega\left(\frac{k}{\varepsilon^{2+z}}\right)$ lower bound for the size of an offline coreset.

## 1 Introduction

Clustering is a fundamental procedure frequently used to help extract important structural information from large datasets. Informally, the goal of clustering is to partition the data into $k$ clusters so that the elements within each cluster have similar properties. Classic formulations of clustering include the $k$-median and $k$-means problems, which have been studied since the 1950's [60, 50]. More generally, for a set $X$ of $n$ points in $\mathbb{R}^d$, along with a metric dist, a cluster parameter $k > 0$, and an exponent $z > 0$ that is a positive integer, the clustering objective can be defined by

$$\min_{C \subset \mathbb{R}^d, |C|=k} \sum_{i=1}^{n} \min_{c \in C} \text{dist}(x_i, c)^z.$$

When dist is the Euclidean distance, the problem is known as $(k, z)$-clustering and more specifically, $k$-median clustering and $k$-means clustering, when $z$ is additionally set to 1 and 2, respectively.

As modern datasets have significantly increased in size, attention has shifted to large-scale computational models, such as the streaming model of computation, that do not require multiple passes over the data. In the (insertion-only) streaming model, the points $x_1, \ldots, x_n$ of $X$ arrive sequentially,

37th Conference on Neural Information Processing Systems (NeurIPS 2023).

and the goal is to output an optimal or near-optimal clustering of $X$ while using space sublinear in $n$, ideally space $k \operatorname{polylog}(n, d)$, since outputting the cluster centers uses $k$ words of space, where each word of space is assumed to be able to store an entire input point in $\mathbb{R}^d$. There exist slight variants of the insertion-only streaming model and a long line of active research has been conducted on clustering in these models [42, 21, 44, 43, 23, 17, 37, 39, 1, 11, 59, 46, 14, 27, 10, 25, 61, 29].

**The sliding window model.** Unfortunately, an important shortcoming of the streaming model is that it ignores the time at which a specific data point arrives and thus it is unable to prioritize recent data over older data. Consequently, the streaming model cannot capture applications in which recent data is more accurate and therefore considered more important than data that arrived prior to a certain time, e.g., Census data or financial markets. Indeed, it has been shown that for a number of applications, the streaming model has inferior performance [4, 52, 57, 62] compared to the sliding window model [33], where only the most recent $W$ updates in the stream comprise the underlying dataset. Here, $W > 0$ is a parameter that designates the window size of the active data, so that all updates before the $W$ most recent updates are considered expired, and the goal is to aggregate statistics about the active data using space sublinear in $W$. In the setting of clustering, where the data stream is $x_1, \ldots, x_n \subset \mathbb{R}^d$, the active data set is $X = \{x_{n-W+1}, \ldots, x_n\}$ for $n \geq W$ and $X = \{x_1, \ldots, x_n\}$ otherwise. Thus the sliding window model is a generalization of the streaming model, depending on the choice of $W$, and is especially relevant for time-sensitive settings, such as data summarization [22, 34], event detection in social media [56], and network monitoring [32, 31, 30].

The sliding window model is especially relevant for applications in which computation *must* be restricted to data that arrived after a certain time. Data privacy laws such as the General Data Protection Regulation (GDPR) mandate that companies cannot retain specific user data beyond a certain duration. For example, the Facebook data policy [36] states that user search histories are retained for 6 months, the Apple differential privacy overview [3] states that collected user information is retained for 3 months, and the Google data retention policy states that browser information may be stored for up to 9 months [41]. These retention polices can be modeled by the sliding window model with the corresponding setting of the window parameter $W$ and thus the sliding window model has been subsequently studied in a wide range of applications [48, 49, 18, 19, 12, 13, 9, 20, 63, 2, 47, 7].

**Clustering in the sliding window model.** Because the clustering objective is not well-suited to popular frameworks such as the exponential histogram or the smooth histogram, there has been significant interest in clustering in the sliding window model. We now describe the landscape of clustering algorithms in the sliding window model; these results are summarized in Table 1. In 2003, [5] first gave a $2^{O(1/\varepsilon)}$-approximation algorithm for $k$-median clustering in the sliding window model using $O\left(\frac{k}{\varepsilon^4} W^{2\varepsilon} \log^2 W\right)$ words of space, where $\varepsilon \in \left(0, \frac{1}{2}\right)$ is an input parameter. Subsequently, [15] gave an $O(1)$-approximate bicriteria algorithm using $2k$ centers and $k^2 \operatorname{polylog}(W)$ space for the $k$-median problem in the sliding window model. The question of whether there exists a $\operatorname{poly}(k \log W)$ space algorithm for $k$-clustering on sliding windows remained open until [16] gave constant-factor approximation sliding window algorithms for $k$-median and $k$-means using $O\left(k^3 \log^6 W\right)$ space and [28] gave constant-factor approximation algorithms for $k$-center clustering using $O\left(k \log \Delta\right)$ space, where $\Delta$ is the aspect ratio, i.e., the ratio of the largest to smallest distances between any pair of points. Afterwards, [8] gave a $C$-approximation algorithm for some constant $C > 2^{14}$, though it should be noted that their main contribution was the first constant-factor approximation algorithm for $k$-clustering using space linear in $k$, i.e., $k \operatorname{polylog}(W, \Delta)$ space, and thus they did not attempt to optimize the constant $C$. Recently, [35] gave the first $(1 + \varepsilon)$-approximation algorithm for $(k, z)$-clustering using $\frac{(kd + d^C)}{\varepsilon^3} \operatorname{polylog}\left(W, \Delta, \frac{1}{\varepsilon}\right)$ words of space, for some constant $C \geq 7$. Using known dimensionality reduction techniques, i.e., [51], the algorithm's dependence on $d^C$ can be removed in exchange for a $\frac{1}{\varepsilon^{14}} \operatorname{polylog}\left(W, \frac{1}{\varepsilon}\right)$ overhead. However, neither the $d^C$ dependency nor the $\frac{1}{\varepsilon^{14}} \operatorname{polylog}\left(W, \frac{1}{\varepsilon}\right)$ trade-off is desirable for realistic settings of $d$ and $\varepsilon$ for applications of $k$-clustering on sliding windows. In particular, recent results have achieved efficient summarizations, i.e., coresets, for $k$-median and $k$-means clustering in the offline setting using $\tilde{O}\left(\frac{k}{\varepsilon^4} \log n\right)$ words of space [27, 25] when the input is from $[\Delta]^d$ and it is known that this is near-optimal, i.e., $\Omega\left(\frac{k}{\varepsilon^{2+z}} \log n\right)$ samples are necessary to form coresets for $(k, z)$-clustering [45] in that setting. Thus a natural question is to ask whether such near-optimal space bounds can be achieved in the sliding window model.

## 1.1 Our Contributions

In this paper, we answer the question in the affirmative. That is, we give near-optimal space algorithms for $k$-median and $k$-means clustering in the sliding window model. In fact, we give more general algorithms for $(k, z)$-clustering in the sliding window that nearly match the space used by the offline coreset constructions of [27, 25, 26]:

**Theorem 1.1.** *There exists an algorithm that samples $\frac{k}{\min(\varepsilon^4, \varepsilon^{2+z})}$ polylog $\frac{n\Delta}{\varepsilon}$ points and with high probability, outputs a $(1 + \varepsilon)$-approximation to $(k, z)$-clustering for the Euclidean distance on $[\Delta]^d$ in the sliding window model.*

In particular, our bounds in Theorem 1.1 achieve $\frac{k}{\varepsilon^4}$ polylog $\frac{n\Delta}{\varepsilon}$ words of space for $k$-median clustering and $k$-means clustering, i.e., $z = 1$ and $z = 2$, respectively, matching the lower bounds of [25, 45] up to polylogarithmic factors.

| Reference | Accuracy | Space | Setting |
|-----------|----------|-------|---------|
| [5] | $2^{O(1/\varepsilon)}$ | $O\left(\frac{k}{\varepsilon^4} W^{2\varepsilon} \log^2 W\right)$ | $k$-median, $\varepsilon \in \left(0, \frac{1}{2}\right)$ |
| [16] | $C > 2$ | $O\left(k^3 \log^6 W\right)$ | $k$-median and $k$-means |
| [34] | $C > 2^{14}$ | $k$ polylog$(W, \Delta)$ | $(k, z)$-clustering |
| [35] | $(1 + \varepsilon)$ | $\frac{(kd + d^{Cz})}{\varepsilon^3}$ polylog $\left(W, \Delta, \frac{1}{\varepsilon}\right), C \geq 7$ | $(k, z)$-clustering |
| Our work | $(1 + \varepsilon)$ | $\frac{k}{\min(\varepsilon^4, \varepsilon^{2+z})}$ polylog$\frac{n\Delta}{\varepsilon}$ | $(k, z)$-clustering |

Table 1: Summary of $(k, z)$-clustering results in the sliding window model for input points in $[\Delta]^d$ on a window of size $W$

Moreover, our algorithm actually produces a coreset, i.e., a data structure that approximately answers the clustering cost of the underlying dataset with respect to any set of $k$ centers, not just the optimal $k$ centers.

**Theorem 1.2.** *There exists an algorithm that samples $\frac{k}{\min(\varepsilon^4, \varepsilon^{2+z})}$ polylog $\frac{n\Delta}{\varepsilon}$ points and with high probability, outputs a $(1 + \varepsilon)$-coreset to $(k, z)$-clustering in the sliding window model for general metrics on $[\Delta]^d$.*

We emphasize that the guarantees of Theorem 1.2 are for general metrics on $[\Delta]^d$, such as $L_p$ metrics. Note that in light of the properties of coresets, the guarantee of Theorem 1.1 follows from taking a coreset for $(k, z)$-clustering on Euclidean distances and then using an offline algorithm for $(k, z)$-clustering for post-processing after the data stream, i.e., see Theorem 2.4.

Along the way, we provide a construction for a $(1+\varepsilon)$-online coreset for $(k, z)$-clustering for general metrics on $[\Delta]^d$. An online coreset for $(k, z)$-clustering is a data structure on a data stream that will not only approximately answer the clustering cost of the underlying dataset with respect to any set of $k$ centers, but also approximately answer the clustering cost of *any prefix of the data stream* with respect to any set of $k$ centers.

**Theorem 1.3.** *There exists an algorithm that samples $\frac{k}{\min(\varepsilon^4, \varepsilon^{2+z})}$ polylog $\frac{n\Delta}{\varepsilon}$ points and with high probability, outputs a $(1 + \varepsilon)$-online coreset for $(k, z)$-clustering.*

We remark that Theorem 1.3 further has the attractive property that once a point is sampled into the online coreset at some point in the stream, then the point irrevocably remains in the online coreset. That is, the online coreset essentially satisfies two different definitions of online: 1) the data structure is a coreset for any prefix of the stream and 2) points sampled into the data structure will never be deleted from the data structure.

We further remark that due to leveraging the coreset construction of [27, 25, 26], we can similarly trade a factor of $\frac{1}{\varepsilon^z}$ for a poly$(k)$ in the guarantees of Theorem 1.1, Theorem 1.2, and Theorem 1.3.

By contrast, the lower bound by [25] states that any offline coreset construction for $k$-means clustering only requires $\Omega\left(\frac{k}{\varepsilon^2}\right)$ points. This lower bound was later strengthened to $\Omega\left(\frac{k}{\varepsilon^{2+z}}\right)$ points by [45], for which matching upper bounds are given by [27, 25]. Thus our online coreset constructions are near-optimal in the $k$ and $\frac{1}{\varepsilon}$ dependencies for $z > 1$ and nearly match the best known offline constructions for $z = 1$.

It is thus a natural question to ask whether our polylogarithmic overheads in Theorem 1.3 are necessary for an $(1 + \varepsilon)$-online coreset. We show that in fact, a logarithmic overhead is indeed necessary to maintain a $(1 + \varepsilon)$-online coreset.

**Theorem 1.4.** *Let $\varepsilon \in (0, 1)$. For sufficiently large $n$, $d$, and $\Delta$, there exists a set $X \subset [\Delta]^d$ of $n$ points $x_1, \ldots, x_n$ such that any $(1 + \varepsilon)$-online coreset for $k$-means clustering on $X$ requires $\Omega\left(\frac{k}{\varepsilon^2} \log n\right)$ points.*

We emphasize that combined with existing offline coreset constructions [25, 26], Theorem 1.4 shows a separation between the problems of constructing offline coresets and online coresets. That is, the problem of maintaining a data structure that recovers coresets for all prefixes of the stream is provably harder than maintaining a coreset for an offline set of points.

## 1.2 Technical Overview

In this section, we give a high-level overview of our techniques. We also describe the limitations of many natural approaches.

**Shortcomings of histograms and sensitivity sampling.** A first attempt at clustering in the sliding window model might be to adapt the popular exponential histogram [33] and smooth histogram techniques [18]. These frameworks convert streaming algorithms to sliding window algorithms in the case that the objective function is smooth, which informally means that once a suffix of a data stream becomes a good approximation of the overall data stream, then it always remains a good approximation, regardless of the values of new elements that arrive in the stream. Unfortunately, [16] showed that the $k$-clustering objective function is not smooth and thus these histogram-based frameworks cannot work. Nevertheless, they gave the first constant-factor approximation by showing that the $k$-clustering objective function is almost-smooth using a generalized triangle inequality, which inherently loses constant factors and thus will not suffice for our goal of achieving a $(1 + \varepsilon)$-approximation.

Another approach might be to adapt the popular sensitivity sampling framework of coreset construction [37, 39, 10, 29]. The sensitivity sampling framework assigns a value to each point, called the sensitivity, which intuitively quantifies the "importance" of that point, and then samples each point with probability proportional to its sensitivity. [9] observed that sliding window algorithms can be achieved from *online* sensitivity sampling, where the importance of each point is measured against the prefix of the stream, and then running the process in reverse at each time, so that more emphasis is placed on the suffix of the sliding window. At a high level, this is the intuition taken by [34, 35], which leverage data structures that prioritize more recent elements of the data stream. However, it is not known how to achieve optimal bounds simply using sensitivity sampling, and indeed the optimal coreset constructions use slightly more nuanced sampling schemes [27, 25].

**Sliding window algorithms from online coresets.** Instead, we recall an observation by [9], who noted that deterministic constructions for online coresets for linear algebraic problems can be utilized to obtain sliding window algorithms for the corresponding linear algebraic problems. We first extend this observation to randomized constructions for online coresets for $k$-clustering problem.

The intuition is quite simple. Given an $(1 + \varepsilon)$-online coreset algorithm for a $k$-clustering problem on a data stream of length $n$ from $\mathbb{R}^d$ that stores $S(n, d, k, \varepsilon, \delta)$ weights points and succeeds with probability $1 - \delta$, we store the $S(n, d, k, \varepsilon', \delta')$ most recent points in the stream, where $\varepsilon' = O\left(\frac{\varepsilon}{\log n}\right)$ and $\delta' = \frac{\delta}{\text{poly}(n)}$. We then feed the $S(n, d, k, \varepsilon', \delta')$ points to the online coreset construction in *reverse order of their arrival*. Since the online coreset preserves all costs for all prefixes of its input, then the resulting data structure will preserve all costs for all *suffixes* of the data stream. To extend this guarantee to the entire stream, including the sliding window, we can then use a standard merge-and-reduce framework. It thus remains to devise a $(1 + \varepsilon)$-online coreset construction for $k$-clustering with near-optimal sampling complexity.

**Online coreset construction.** To that end, our options are quite limited, as to the best of our knowledge, the only offline coreset constructions using $\tilde{O}\left(\frac{k}{\varepsilon^4} \log n\right)$ words of space when the input is from $[\Delta]^d$ are due to [27, 25]. Fortunately, although the analyses of correctness for these sampling

schemes are quite involved, the constructions themselves are quite accessible. For example, [27] first uses an $(\alpha, \beta)$-approximation, i.e., a clustering that achieves $\alpha$-approximation to the optimal cost but uses $\beta k$ centers, to partition the underlying dataset $X$ into disjoint concentric rings around each of the $\beta k$ centers. These rings are then gathered into groups and it is shown that by independently sampling a fixed number of points with replacement from each of the groups suffices to achieve a $(1 + \varepsilon)$-coreset. Their analysis argues that the contribution of each of the groups toward the overall $k$-clustering cost is preserved through an expectation and variance bounding argument, and then taking a sophisticated union bound over a net over the set of possible centers. Thus their argument still holds when each point of the dataset is independently sampled by the data structure with probability proportional to the probability it would have been sampled by the group. Moreover, independently sampling each point with a higher probability can only decrease the variance, so that correctness is retained, though we must also upper bound the number of sampled points. Crucially, independently sampling each point can be implemented in the online setting and the probability of correctness can be boosted to union bound over all times in the stream, which facilitates the construction of our $(1 + \varepsilon)$-online coreset, given an $(\alpha, \beta)$-approximation.

**Consistent $(\alpha, \beta)$-approximation.** It seemingly remains to find $(\alpha, \beta)$-approximations for $k$-clustering at all times in the stream. A natural approach would be to use an algorithm that achieves a $(\alpha, \beta)$-approximation at a certain time in the stream with constant probability, e.g., [59], boost the probability of success to $1 - \frac{1}{\text{poly}(n)}$, and the union bound to argue correctness over all times in the stream. However, a subtle pitfall here is that the rings and groups in the offline coreset construction of [27] are with respect to a specific $(\alpha, \beta)$-approximation. Hence their analysis would no longer hold if a point $x_t$ was assigned to cluster $i_1$ at time $t$ when the sampling process occurs but then assigned to cluster $i_2$ at the end of the stream. Therefore, we require a consistent $(\alpha, \beta)$-approximation, so that once the algorithm assigns point $x_t$ to cluster $i$, then the point $x_t$ will always remain in cluster $i$ even if a newer and closer center is subsequently opened later in the stream. To that end, we invoke a result of [34] that analyzes the popular Meyerson online facility location algorithm, along with a standard guess-and-double approach for estimating the input parameter to the Meyerson subroutine.

**Lower bound.** The intuition for our lower bound that any $(1+\varepsilon)$-online coreset for $(k, z)$-clustering requires $\Omega\left(\frac{k}{\varepsilon^2}\right)$ is somewhat straightforward and in a black-box manner. We first observe that [25] showed the existence of a set $X$ of $\Omega\left(\frac{k}{\varepsilon^2}\right)$ unit vectors in $\mathbb{R}^d$ such that any coreset with $o\left(\frac{k}{\varepsilon^2}\right)$ samples provably cannot accurately estimate the $(k, z)$-clustering cost for a set $C$ of $k$ unit vectors.

Since an online $(1 + \varepsilon)$-coreset must answer queries on all prefixes of the stream, we embed $\Omega(\log n)$ instances of $X$. We first increase the dimension by a $\log n$ factor so that each of these instances can have disjoint support. We then give each of the instances increasingly exponential weight to force the data structure to sample $\Omega\left(\frac{k}{\varepsilon^2}\right)$ points for each instance. Specifically, we insert $\tau^i$ copies of the $i$-th instance of $X$, where $\tau > 1$ is some constant. Because the weight of the $i$-th instance is substantially greater than the sum of the weights of all previous instances, then any $(1 + \varepsilon)$-online coreset must essentially be a $(1 + \varepsilon)$-offline coreset for the $i$-th instance, thus requiring $\Omega\left(\frac{k}{\varepsilon^2}\right)$ points for the $i$-th instance. This reasoning extends to all $\Omega(\log n)$ instances, thus showing that any online $(1 + \varepsilon)$-coreset requires $\Omega\left(\frac{k}{\varepsilon^2} \log n\right)$ points.

## 2 Algorithm

In this section, we describe our sliding window algorithm for $k$-clustering. We first overview the construction of an online $(1 + \varepsilon)$ coreset for $(k, z)$-clustering under general discrete metrics. We then describe how our online coreset construction for $(k, z)$-clustering on general discrete metric spaces can be used to achieve near-optimal space algorithms for $(k, z)$-clustering in the sliding window model.

**Online $(1 + \varepsilon)$-coreset.** We first recall the following properties from the Meyerson sketch, which we formally introduce in Appendix A.

**Theorem 2.1.** *[8] Given an input stream $x_1, \ldots, x_n \in \mathbb{R}^d$ defining a set $X \subset [\Delta]^d$, there exists an online algorithm* MULTMEYERSON *that with probability at least $1 - \frac{1}{\text{poly}(n)}$:*

*(1) on the arrival of each point $x_i$, assigns $x_i$ to a center in $C$ through a mapping $\pi : X \to C$, where $C$ contains at most $O\left(2^{2z}k \log n \log \Delta\right)$ centers*

*(2) $\sum_{x \in X} \|x_i - \pi(x_i)\|_2^z \leq 2^{z+7} \operatorname{Cost}_{|S| \leq k}(X, S)$*

*(3) MULTMEYERSON uses $O\left(2^z k \log^3(nd\Delta)\right)$ words of space*

We also use the following notation, adapted from [27] to the online setting.

Let $\mathcal{A}$ be an $(\alpha, \beta)$-approximation for a $k$-means clustering on an input set $X \subseteq [\Delta]^d$ and let $C_1, \ldots, C_{\beta k}$ be the clusters of $X$ induced by $\mathcal{A}$. Suppose the points of $X$ arrive in a data stream $S$. For a fixed $\varepsilon > 0$, define the following notions of rings and groups:

- The average cost of cluster $C_i$ is denoted by $\kappa_{C_i} := \frac{\operatorname{Cost}(C_i, \mathcal{A})}{|C_i|}$.

- For any $i, j$, the ring $R_{i,j}$ is the set of points $x \in C_i$ such that $2^j \kappa_{C_i} \leq \operatorname{Cost}(x, \mathcal{A}) < 2^{j+1} \kappa_{C_i}$. For any $j$, $R_j = \cup R_{i,j}$.

- The inner ring $R_I(C_i) = \cup_{j \leq 2z \log \frac{\varepsilon}{2}} R_{i,j}$ is the set of points of $C_i$ with cost at most $\left(\frac{\varepsilon}{z}\right)^{2z} \kappa_{C_i}$. More generally for a solution $\mathcal{S}$, let $R_I^{\mathcal{S}}$ denote the union of the inner rings induced by $\mathcal{S}$.

- The outer ring $R_O(C_i) = \cup_{j \geq 2z \log \frac{z}{\varepsilon}} R_{i,j}$ is the set of points of $C_i$ with cost at least $\left(\frac{z}{\varepsilon}\right)^{2z} \kappa_{C_i}$. More generally for a solution $\mathcal{S}$, let $R_O^{\mathcal{S}}$ denote the union of the outer rings induced by $\mathcal{S}$.

- The main ring $R_M(C_i)$ is the set of points of $C_i$ that are not in the inner or outer rings, i.e., $R_M(C_i) = C_i \setminus (R_I(C_i) \cup R_O(C_i))$.

- For any $j$, the group $G_{j,b}$ consists of the $(2^{b-1} + 1)$-th to $(2^b)$-th points of each ring $R_{i,j}$ that arrive in $S$.

- For any $j$, we use $G_{j,\min}$ to denote the union of the groups with the smallest costs, i.e.,

$$G_{j,\min} = \left\{ x \mid \exists i, x \in R_{i,j}, \operatorname{Cost}(R_{i,j}, \mathcal{A}) < 2\left(\frac{\varepsilon}{4z}\right)^z \frac{\operatorname{Cost}(R_j, \mathcal{A})}{\beta k} \right\}.$$

- The outer groups $G_b^O$ partition the outer rings $R_O^{\mathcal{A}}$ so that

$$G_b^O = \left\{ x \mid \exists i, x \in C_i, \left(\frac{\varepsilon}{4z}\right)^z \frac{\operatorname{Cost}(R_O^{\mathcal{A}}, \mathcal{A})}{\beta k} \cdot 2^b \leq \operatorname{Cost}(R_O(C_i), \mathcal{A}) < \left(\frac{\varepsilon}{4z}\right)^z \frac{\operatorname{Cost}(R_O^{\mathcal{A}}, \mathcal{A})}{\beta k} \cdot 2^{b+1} \right\}.$$

- We define $G_{\min}^O = \cup_{b \leq 0} G_b^O$ and $G_{\max}^O = \cup_{b \geq z \log \frac{4z}{\varepsilon}} G_b^O$.

---

**Algorithm 1** RINGSAMPLE

---

**Input:** Points $x_1, \ldots, x_n \in [\Delta]^d$
**Output:** A set $W$ of weighted points and timestamps
1: Initiate an instance of $(\alpha, \beta)$-bicriteria algorithm MULTMEYERSON
2: $\gamma \leftarrow \frac{C \max(\alpha^2, \alpha^z)\beta}{\min(\varepsilon^2, \varepsilon^z)} \log^2 \frac{1}{\varepsilon} \left(k \log |\mathbb{C}| + \log\log \frac{1}{\varepsilon} + \log n\right) \log^2 \frac{1}{\varepsilon}$
3: $W \leftarrow \emptyset$
4: **for** each point $x_t$, $t \in [n]$ **do**
5:     Let $c_i$ be the center assigned for $x_t$ by MULTMEYERSON
6:     Let $2^j \leq \|x_t - c_i\|_2^z < 2^{j+1}$ for $j \in \mathbb{Z}$
7:     Let $b \in \mathbb{Z}$ so that the number of points in $R_{i,j}$ is between $2^{b-1} + 1$ and $2^b$
8:     Let $r_t$ be the number of points in $G_{j,b}$ at time $t$
9:     $p_x \leftarrow \min\left(\frac{4}{r_t} \cdot \gamma \log n, 1\right)$
10:     With probability $p_x$, add $x$ to $W$ with timestamp $t$ and weight $\frac{1}{p_x}$
11: **return** $W$

---

We then adapt the offline coreset construction of [27] to an online setting at the cost of logarithmic overheads, which suffice for our purpose. The algorithm (Algorithm 1) has the following guarantees:

**Lemma 2.2.** *Let $\mathbb{C}$ be an $\mathcal{A}$-approximate centroid set for a fixed group $G$. There exists an algorithm* RINGSAMPLE *that samples*

$$O\left(\frac{\max(\alpha^2, \alpha^z)\beta}{\min(\varepsilon^2, \varepsilon^z)} \log^2 \frac{1}{\varepsilon} \left(k \log |\mathbb{C}| + \log\log \frac{1}{\varepsilon} + \log n\right) \log^2 n \log^2 \Delta \log^2 \frac{1}{\varepsilon}\right)$$

*points and with high probability, outputs a $(1+\varepsilon)$-online coreset for the $k$-means clustering problem.*

Informally, an approximate centroid set is a set of possible points so that taking the centers from this set generates an approximately accurate solution (see Appendix B for a formal definition). To bound $\log |\mathbb{C}|$, we construct and apply a terminal embedding to project each point to a lower dimension and then appeal to known bounds for approximate centroid sets in low-dimensional Euclidean, thereby giving our online coreset algorithm with the guarantees of Theorem 1.3.

**Sliding window model.** We first recall a standard approach for using offline coreset constructions for insertion-only streaming algorithms. Suppose there exists a randomized algorithm that produces an online coreset algorithm that uses $S(n, \varepsilon, \delta)$ points for an input stream of length $n$, accuracy $\varepsilon$, and failure probability $\delta$, where for the ease of discussion, we omit additional dependencies. A standard approach for using coresets on insertion-only streams is the merge-and-reduce approach, which partitions the stream into blocks of size $S\left(n, \frac{\varepsilon}{2\log n}, \frac{\delta}{\text{poly}(n)}\right)$ and builds a coreset for each block. Each coreset is then viewed as the leaves of a binary tree with height at most $\log n$, since the binary tree has at most $n$ leaves. Then at each level of the binary tree, for each node in the level, a coreset of size $S\left(n, \frac{\varepsilon}{2\log n}, \frac{\delta}{\text{poly}(n)}\right)$ is built from the coresets representing the two children of the node. Due to the mergeability property of coresets, the coreset at the root of the tree will be a coreset for the entire stream with accuracy $\left(1 + \frac{\varepsilon}{2\log n}\right)^{\log n} \le (1 + \varepsilon)$ and failure probability $\delta$.

This approach fails for sliding window algorithms because the elements at the beginning of the data stream can expire, and so coresets corresponding to earlier blocks of the stream may no longer accurate, which would result in the coreset at the root of the tree also no longer being accurate. On the other hand, suppose we partition the stream into blocks consisting of $S\left(n, \frac{\varepsilon}{2\log n}, \frac{\delta}{\text{poly}(n)}\right)$ elements as before, but instead of creating an offline coreset for each block, we can create an online coreset for the elements *in reverse*. That is, since the elements in each block are explicitly stored, we can create offline an artificial stream consisting of the elements in the block in reverse and then give the artificial stream as input to the online coreset construction. Note that if we also first consider the "latter" coreset when merging two coresets, then this effectively reverses the stream. Moreover, by the correctness of the online coreset, our data structure provides correctness over any prefix of the reversed stream, or equivalently, any suffix of the stream and specifically, correctness over the sliding window. We thus further adapt the merge-and-reduce framework to show that randomized online coresets for problems in clustering can also be used to achieve randomized algorithms for the corresponding problems in the sliding window model. We formalize this approach in Algorithm 2.

**Theorem 2.3.** *Let $x_1, \ldots, x_n$ be a stream of points in $[\Delta]^d$, $\varepsilon > 0$, and let $X = \{x_{n-W+1}, \ldots, x_n\}$ be the $W$ most recent points. Suppose there exists a randomized algorithm that with probability at least $1 - \delta$, outputs an online coreset algorithm for a $k$-clustering problem with $S(n, d, k, \varepsilon, \delta)$ points. Then there exists a randomized algorithm that with probability at least $1 - \delta$, outputs a coreset for the $k$-clustering problem in the sliding window model with $O\left(S\left(n, d, k, \frac{\varepsilon}{\log n}, \frac{\delta}{n^2}\right) \log n\right)$ points.*

By Theorem 1.3 and Theorem 2.3, we have:

**Theorem 2.4.** *There exists an algorithm that samples $\frac{k}{\min(\varepsilon^4, \varepsilon^{2+z})} \text{polylog} \frac{n\Delta}{\varepsilon}$ points and with high probability, outputs a $(1+\varepsilon)$-coreset to $(k, z)$-clustering in the sliding window model.*

Using an offline algorithm for $(k, z)$-clustering for post-processing after the data stream, we have Theorem 1.1.

## 3  Experimental Evaluations

In this section, we conduct simple empirical demonstrations as proof-of-concepts to illustrate the benefits of our algorithm. Our empirical evaluations were conducted using Python 3.10 using a 64-bit

---

**Algorithm 2** Merge-and-reduce framework for randomized algorithms in the sliding window model, using randomized constructions of online coresets

---

**Input:** A clustering function $f$, a set of points $x_1, \ldots, x_n \subseteq \mathbb{R}^d$, accuracy parameter $\varepsilon > 0$, failure probability $\delta \in (0, 1)$, and window size $W > 0$

**Output:** An approximation of $f$ on the $W$ most recent points

1: Let $\text{CORESET}(X, n, d, k, \varepsilon, \delta)$ be an online coreset construction with $S(n, d, k, \varepsilon, \delta)$ points on a set $X \subseteq \mathbb{R}^d$

2: $m \leftarrow O\left(S\left(n, d, k, \frac{\varepsilon}{\log n}, \frac{\delta}{n}\right) \log n\right)$

3: Initialize blocks $B_0, B_1, \ldots, B_{\log n} \leftarrow \emptyset$

4: **for** each point $x_t$ with $t \in [n]$ **do**

5:      **if** $B_0$ does not contain $m$ points **then**

6:          Prepend $x_t$ to $B_0$, i.e., $B_0 \leftarrow \{x_t\} \cup B_0$

7:      **else**

8:          Let $i$ be the smallest index such that $B_i = \emptyset$

9:          $B_i \leftarrow \text{CORESET}\left(Y, n, d, k, \frac{\varepsilon}{\log n}, \frac{\delta}{n^2}\right)$ for $Y = B_0 \cup \ldots \cup B_{i-1}$   ▷$Y$ is an ordered set of weighted points

10:          **for** $j = 0$ to $j = i - 1$ **do**

11:              $B_j \leftarrow \emptyset$

12:          $B_0 \leftarrow \{x_t\}$

13:      **return** the ordered set $B_{\log n} \cup \ldots \cup B_0$

---

operating system on an AMD Ryzen 7 5700U CPU, with 8GB RAM and 8 cores with base clock 1.80 GHz. The general approach to our experiments is to produce a data stream $S$ that defines dataset $X$, whose generation we describe below, as well as in Appendix F. We then compare the performance of a simplified version of our algorithm with various state-of-the-art baselines.

**Baselines.** Our first baseline (denoted off for offline) is the simple Lloyd's algorithm on the entire dataset $X$, with multiple iterations using the k-means++ initialization. This is a standard approach for finding a good approximation to the optimal clustering cost, because finding the true optimal centers requires exponential time. Because this offline Lloyd's algorithm has access to the entire dataset, the expected behavior is that this algorithm will have the best objective, i.e., smallest clustering cost. However, we emphasize that this algorithm requires storing the entire dataset $X$ in memory and thus its input size is significantly larger than the sublinear space algorithms.

To compare with the offline Lloyd's algorithm, we run a number of sublinear space algorithms. These algorithms generally perform some sort of processing on the datastream $X$ to create a coreset $C$. We normalize the space requirement of these algorithms by permitting each algorithm to store $m$ points across specific ranges of $m$. We then run Lloyd's algorithm on the coreset $C$, with the same number of iterations using the k-means++ initialization.

Our first sublinear space algorithm is uniform sampling on the dataset $X$. That is, we form $C$ by uniformly sampling $m$ points from $X$, before running Lloyd's algorithm. We use uni to denote this algorithm whose first step is based on uniformly sampling. Our second sublinear space algorithm is the importance sampling approach used by histogram-based algorithms, e.g., [15, 11, 8]. These algorithms perform importance sampling, i.e., sample points into the coreset $C$ with probability proportional to their distances from existing samples and delete points once the clustering cost of $C$ is much higher than the clustering cost of the dataset $X$. We use hist(ogram) to denote this algorithm that is based on the histogram frameworks for sliding windows.

Our final algorithm is a simplification of our algorithm. As with the histogram-based algorithm, we perform importance sampling on the stream $S$ to create the coreset $C$ of size $m$. Thus we do not implement the ring and group sampling subroutines in our full algorithm. However, the crucial difference compared to the histogram-based approach is that we forcefully discard points of $C$ that have expired. We use imp to denote this algorithm whose first step is based on importance sampling.

**Dataset.** We first describe the methodology and experimental setup of our empirical evaluation on a real-world dataset with an amount of synthetic noise before detailing the experimental results. The

first component of our dataset consists of the points of the SKIN (Skin Segmentation) dataset $X'$ from the publicly available UCI repository [6], which was also used in the experiments of [8]. The dataset $X'$ consists of $245,057$ points with four features, where each point refers to a separate image, such that the first three features are constructed over BGR space, and the fourth feature is the label for whether or not the image refers to a skin sample. We subsequently pre-process each dataset to have zero mean and unit standard deviation in each dimension.

We then form our dataset $X$ by augmenting $X'$ with 201 points in four-dimensional space, where 100 of these points were drawn from a spherical Gaussian with unit standard deviation in each direction and centered at $(-10, 10, 0, 0)$ and 100 of these points were drawn from a spherical Gaussian with unit standard deviation in each direction and centered at $(10, -10, 0, 0)$. The final point of $X$ was drawn from a spherical Gaussian with unit standard deviation centered at $(500, 500, 0, 0)$. Thus our dataset $X$ has dimensions $n = 245,258$ and $d = 4$. We then create the data stream $S$ by prepending two additional points drawn from spherical Gaussians with standard deviation 2.75 centered at $(-10, 10, 0, 0)$ and $(-10, -10, 0, 0)$ respectively, so that the stream has length $245,260$. We set the window length to be $245,258$ in accordance with the "true" data set, so that the first two points of the stream will be expired by the data stream.

**Experimental setup.**    For each of the instances of Lloyd's algorithm, either on the entire dataset $X$ or the sampled coreset $C$, we use 10 iterations using the k-means++ initialization. While the offline Lloyd's algorithm stores the entire dataset $X$ of $245,258$ points in memory, we only allow each of the sublinear-space algorithms to store a fixed $m$ points. We compare the algorithms across $m \in \{5, 10, 15, 20, 25, 30\}$ and $k \in \{2, 3, 4, 5, 6, 7, 8, 9, 10\}$. Note that in the original dataset, each of the points has a label for either skin or non-skin, which would be reasonable for $k = 2$. However, due to the artificial structure possibly induced by the synthetic noise, it also makes sense to other values of $k$. In particular, preliminary experiments from uniform sampling by the elbow method indicated that $k = 3$ would be a reasonable setting. Thus we fix $k = 3$ while varying $m \in \{5, 10, 15, 20, 25, 30\}$ and we arbitrarily fix $m = 25$ while varying $k \in \{2, 3, 4, 5, 6, 7, 8, 9, 10\}$.

**Experimental results.**    For each choice of $m$ and $k$, we ran each algorithm 30 times and tracked the resulting clustering cost. Our algorithm demonstrated superior performance than the other sublinear-space algorithms across all values of $m \in \{5, 10, 15, 20, 25, 30\}$ and $k \in \{2, 3, 4, 5, 6, 7, 8, 9, 10\}$, and was even quite competitive with the offline Lloyd's algorithm, even though our algorithm only used memory size $m \le 30$, while the offline algorithm used memory $245,258$.

Uniform sampling performed well for $k = 2$, which in some case captures the structure imposed on the data through the skin vs. non-skin label, but for larger $k$, the optimal solutions start placing centers to handle the synthetic noise, which may not be sampled by uniform sampling. Thus uniform sampling performed relatively poorly albeit quite stably for larger $k$. In contrast, the histogram-based algorithm performed poorly for small $k$ across all our ranges of $m$, due to sampling the extra points in $S \setminus X$, so that the resulting Lloyd's algorithm on $C$ moved the centers far away from the optimal centers of $X$. On the other hand, the histogram-based algorithm performed well for larger $k$, likely due to additional centers that could be afforded to handle the points in $S \setminus X$. We plot our results in Figure 1 and defer additional experiments to Appendix F.

## 4    Conclusion

In this paper, we give an algorithm outputs a $(1 + \varepsilon)$-approximation to $(k, z)$-clustering in the sliding window model, while using $\frac{k}{\min(\varepsilon^4, \varepsilon^{2+z})}$ polylog $\frac{n\Delta}{\varepsilon}$ words of space when the points are from $[\Delta]^d$. Our algorithm not only improves on a line of work [5, 16, 34, 8, 35], but also nearly matches the space used by the offline coreset constructions of [27], which is known to be near-optimal in light of a $\Omega\left(\frac{k}{\varepsilon^{2+z}}\right)$ lower bound for the size of an offline coreset [45].

We also give a lower bound that shows a logarithmic overhead in the number of points is needed to maintain a $(1 + \varepsilon)$-online coreset compared to a $(1 + \varepsilon)$-coreset. That is, we gave in Theorem 1.4 a set $X \subset [\Delta]^d$ of $n$ points $x_1, \ldots, x_n$ such that any $(1 + \varepsilon)$-online coreset for $k$-means clustering on $X$ requires $\Omega\left(\frac{k}{\varepsilon^2} \log n\right)$ points. However, this does not rule out whether the $\log n$ overhead is necessary for $(k, z)$-clustering in the sliding window model, since a sliding window algorithm does

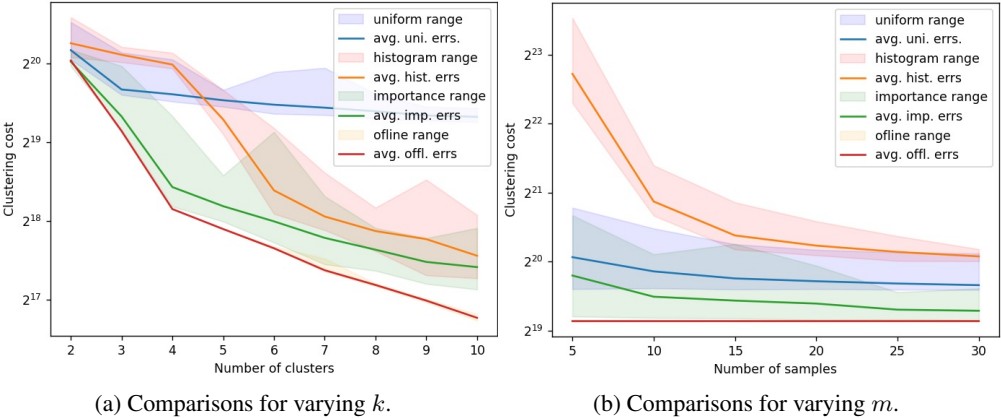

(a) Comparisons for varying $k$.  (b) Comparisons for varying $m$.

Fig. 1: Comparison of average clustering costs made by uniform sampling, histogram-based algorithm, and our coreset-based algorithm across various settings of space allocated to the algorithm, given a synthetic dataset. For comparison, we also include the offline k-means++ algorithm as a baseline, though it is inefficient because it stores the entire dataset.

not necessarily need to maintain an online coreset. We leave this question as a possible direction for future work.

## Acknowledgments

David P. Woodruff and Samson Zhou were partially supported by a Simons Investigator Award and by the National Science Foundation under Grant No. CCF-1815840. This work was done in part while Samson Zhou was at Carnegie Mellon University, UC Berkeley, and Rice University.

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

# A  Preliminaries

For a positive integer $n$, we use the notation $[n]$ to denote the set $\{1, \ldots, n\}$. Similarly, we use $[\Delta]^d$ to denote $\{1, \ldots, \Delta\}^d$. We use $\mathrm{poly}(n)$ to denote a fixed polynomial in $n$ with degree determined as necessary by setting the appropriate constants in corresponding variables. Similarly, we use $\mathrm{polylog}(n)$ to denote $\mathrm{poly}(\log n)$. We suppress polylogarithmic dependencies by writing $\tilde{O}(f(\cdot)) = O(f(\cdot))\,\mathrm{polylog}\, f(\cdot)$.

For $(k, z)$-clustering on a set $X = \{x_1, \ldots, x_n\} \subset \mathbb{R}^d$ using a set $C$ of $k$ centers and a distance function $\mathrm{dist}(\cdot, \cdot)$, we define the notation $\mathrm{Cost}(X, C) = \sum_{i=1}^{n} \min_{c \in C} \mathrm{dist}(x_i, c)^z$. We also define the notation $\mathrm{Cost}_{|S| \le k}(X, S) := \min_{S : |S| \le k} \mathrm{Cost}(X, S)$, so that $\mathrm{Cost}_{|S| \le k}$ is the cost of an optimal $(k, z)$-clustering.

**Definition A.1** (($\alpha, \beta$)-approximation)**.** *We say a set of centers $C$ provides an $(\alpha, \beta)$-approximation to the optimal $k$-means clustering on a set $X$ if $|C| \le \beta k$ and*

$$\mathrm{Cost}(X, C) \le \alpha \mathsf{OPT}.$$

**Definition A.2** (Coreset)**.** *A coreset for $(k, z)$-clustering on an approximation parameter $\varepsilon > 0$ and a set $X$ of points $x_1, \ldots, x_n \in \mathbb{R}^d$ with distance function $\mathrm{dist}$ is a subset $S$ of weighted points of $X$ with weight function $w$ such that for any set $C$ of $k$ points, we have*

$$(1 - \varepsilon) \sum_{i=1}^{n} \mathrm{dist}(x_i, C)^z \le \sum_{q \in S} w(q) \, \mathrm{dist}(q, S)^z \le (1 + \varepsilon) \sum_{i=1}^{n} \mathrm{dist}(x_i, C)^z.$$

**Definition A.3** (Online Coreset)**.** *An online coreset for $(k, z)$-clustering on an approximation parameter $\varepsilon > 0$ and a set $X$ of points $x_1, \ldots, x_n \in \mathbb{R}^d$ with distance function $\mathrm{dist}$ is a subset $S$ of weighted points of $X$ with weight function $w$ such that for any set $C$ of $k$ points and for any $t \in [n]$, we have*

$$(1 - \varepsilon) \sum_{i=1}^{t} \mathrm{dist}(x_i, C)^z \le \sum_{q \in S_t} w(q) \, \mathrm{dist}(q, S_t)^z \le (1 + \varepsilon) \sum_{i=1}^{t} \mathrm{dist}(x_i, C)^z,$$

*where $S_t = S \cap \{X_1, \ldots, X_t\}$, i.e., the subset of $S$ that has arrived at time $t$.*

**Theorem A.4** (Bernstein's inequality)**.** *Let $X_1, \ldots, X_n$ be independent random variables such that $\mathbb{E}[X_i^2] < \infty$ and $X_i \ge 0$ for all $i \in [n]$. Let $X = \sum_i X_i$ and $\gamma > 0$. Then*

$$\mathbf{Pr}\left[X \le \mathbb{E}[X] - \gamma\right] \le \exp\left(\frac{-\gamma^2}{2 \sum_i \mathbb{E}[X_i^2]}\right).$$

*If $X_i - \mathbb{E}[X_i] \le \Delta$ for all $i$, then for $\sigma_i^2 = \mathbb{E}[X_i^2] - \mathbb{E}[X_i]^2$,*

$$\mathbf{Pr}\left[X \ge \mathbb{E}[X] + \gamma\right] \le \exp\left(\frac{-\gamma^2}{2 \sum_i \sigma_i^2 + 2 \gamma \Delta / 3}\right).$$

**Meyerson sketch.** We briefly review the Meyerson sketch [54] and the relevant properties that we need from the Meyerson sketch. The Meyerson sketch provides an $(\alpha, \beta)$-approximation to $(k, z)$-clustering on a data stream of points $x_1, \ldots, x_n \in [\Delta]^d$ with $\alpha = 2^{z+7}$ and $\beta = O\left(2^{2z} \log n \log \Delta\right)$. Moreover, for our purposes, it provides the crucial property that on the arrival of each point $x_i$, the algorithm irrevocably assigns $x_i$ to one of the $\beta k$ centers. Specifically, the clustering cost at the end of the stream is computed with respect to the center that $x_i$ is assigned at time $i$, which may not be the closest center to $x_i$ because the closer center can be opened at a later time.

For the ease of discussion, we describe the Meyerson sketch for $z = 1$; the intuition generalizes naturally to other values of $z$. The Meyerson sketch performs via a guess-and-double approach, where it first attempts to guess the cost of the optimal clustering cost. Using the guess of the cost, it then turns each point into a center with probability proportional to the distance of that point from the existing centers. This subroutine is illustrated in Algorithm 3. If too many centers have been opened, then the Meyerson sketch determines that the guess for the optimal clustering cost must have been too low and increases the guess. The overall algorithm is given in Algorithm 4.

We require the following properties from the Meyerson sketch.

---

**Algorithm 3** High probability $\mathrm{MEYERSON}(X, \widetilde{\mathrm{OPT}}, \alpha, \delta, \Delta, z, k)$ sketch

---

**Input:** Points $X := x_1, \ldots, x_n \in \mathbb{R}^d$ with aspect ratio $\Delta$, estimate $\widetilde{\mathrm{OPT}} \geq 0$ such that $\alpha\mathrm{OPT} \leq$
$\quad$ $\widetilde{\mathrm{OPT}} \leq \mathrm{OPT}$ for some $\alpha \in (0, 1)$, failure probability $\delta \in (0, 1)$
**Output:** A coreset for $k$-clustering on $X$
1: $\gamma \leftarrow 2 \log \frac{1}{\delta}$
2: **for** $i \in [\gamma]$ **do**
3: $\quad$ **for** $t \in [n]$ **do**
4: $\quad\quad$ **if** $t = 1$ **then**
5: $\quad\quad\quad$ $M_i \leftarrow x_1$, $C_{\mu_i} \leftarrow 0$, $w_i(x_1) = 1$
6: $\quad\quad$ **else**
7: $\quad\quad\quad$ **if** $|M_i| \leq 4k(1 + \log \Delta)\left(\frac{2^{z+3}}{\alpha^z} + 1\right)$ **then**
8: $\quad\quad\quad\quad$ With probability $\min\left(\frac{k(1+\log \Delta) \operatorname{dist}(x_t, M_i)^z}{\widetilde{\mathrm{OPT}}}, 1\right)$, add $x_t$ to $M_i$ with weight 1,
$\quad$ i.e., $w_i(x_t) = 1$
9: $\quad\quad\quad\quad$ Otherwise, let $z = \operatorname{argmin}_{y \in M_i} \operatorname{dist}(x_t, y)$, increment the weight of $z$, i.e.,
$\quad$ $w_i(z) \leftarrow w_i(z) + 1$, and increase $C_{\mu_i} \leftarrow C\mu_i \operatorname{dist}(x_t, z)^p$
10: Let $j = \operatorname{argmin}_{i:|M_i| \leq 4k(1+\log \Delta)\left(\frac{2^{z+3}}{\alpha^z}+1\right)} C_{\mu_i}$ be the index of the minimal cost sketch with at

$\quad$ most $4k(1 + \log \Delta)\left(\frac{2^{z+3}}{\alpha^z} + 1\right)$ samples $\quad\quad\quad$ ▷Return FAIL if such $j$ does not exist
11: **return** $\cup_{i \in [\gamma]} M_i$, $w_j$, and $C_{\mu_j}$

---

---

**Algorithm 4** High probability $\mathrm{MULTMEYERSON}$ sketch

---

**Input:** Points $X := x_1, \ldots, x_n \in \mathbb{R}^d$ with aspect ratio $\Delta$, estimate $\widetilde{\mathrm{OPT}} \geq 0$ such that $\alpha\mathrm{OPT} \leq$
$\quad$ $\widetilde{\mathrm{OPT}} \leq \mathrm{OPT}$ for some $\alpha \in (0, 1)$, failure probability $\delta \in (0, 1)$
**Output:** A coreset for $k$-means clustering on $X$ if $\widetilde{\mathrm{OPT}}$ upper bounds the cost of the optimal
$\quad$ clustering
1: $\gamma \leftarrow \log nd(\Delta^z)$
2: **for** $i \in [\gamma]$ **do**
3: $\quad$ Run $\mathrm{MEYERSON}\left(X, 2^i, \alpha = \frac{1}{2}, \delta, \Delta, z, k\right)$ in parallel
4: Let $j$ be the minimal index in $[\gamma]$ such that $\mathrm{MEYERSON}$ with input $2^j$ has size smaller than
$\quad$ $8k \log \frac{1}{\delta}(1 + \log \Delta)\left(2^{2z+3} + 1\right)$ and cost smaller than $2^{z+6+j}$
5: **return** the output for $\mathrm{MEYERSON}\left(X, 2^j, \alpha = \frac{1}{2}, \delta, \Delta, z, k\right)$

---

**Theorem 2.1.** *[8] Given an input stream $x_1, \ldots, x_n \in \mathbb{R}^d$ defining a set $X \subset [\Delta]^d$, there exists an online algorithm $\mathrm{MULTMEYERSON}$ that with probability at least $1 - \frac{1}{\operatorname{poly}(n)}$:*

*(1) on the arrival of each point $x_i$, assigns $x_i$ to a center in $C$ through a mapping $\pi : X \to C$, where $C$ contains at most $O\left(2^{2z} k \log n \log \Delta\right)$ centers*

*(2) $\sum_{x \in X} \|x_i - \pi(x_i)\|_2^z \leq 2^{z+7} \operatorname{Cost}_{|S| \leq k}(X, S)$*

*(3) $\mathrm{MULTMEYERSON}$ uses $O\left(2^z k \log^3(nd\Delta)\right)$ words of space*

# B  Online $(1 + \varepsilon)$-Coreset

In this section, we describe how to construct an online $(1 + \varepsilon)$ coreset for $(k, z)$-clustering under general discrete metrics. We first describe the offline coreset construction of [27] and then argue that the construction can be adapted to an online setting at the cost of logarithmic overheads, which suffice for our purpose.

Let $\mathcal{A}$ be an $(\alpha, \beta)$-approximation for a $(k, z)$-clustering on an input set $X \subseteq [\Delta]^d$ and let $C_1, \ldots, C_{\beta k}$ be the clusters of $X$ induced by $\mathcal{A}$. Suppose the points of $X$ arrive in a data stream $S$. For a fixed $\varepsilon > 0$, [27] define the following notions of rings and groups:

- The average cost of cluster $C_i$ is denoted by $\kappa_{C_i} := \frac{\mathrm{Cost}(C_i, \mathcal{A})}{|C_i|}$.

- For any $i, j$, the ring $R_{i,j}$ is the set of points $x \in C_i$ such that
$$2^j \kappa_{C_i} \le \mathrm{Cost}(x, \mathcal{A}) < 2^{j+1} \kappa_{C_i}.$$
For any $j$, $R_j = \cup R_{i,j}$.

- The inner ring $R_I(C_i) = \cup_{j \le 2z \log \frac{\varepsilon}{z}} R_{i,j}$ is the set of points of $C_i$ with cost at most $\left(\frac{\varepsilon}{z}\right)^{2z} \kappa_{C_i}$. More generally for a solution $\mathcal{S}$, let $R_I^{\mathcal{S}}$ denote the union of the inner rings induced by $\mathcal{S}$.

- The outer ring $R_O(C_i) = \cup_{j \ge 2 \log \frac{z}{\varepsilon}} R_{i,j}$ is the set of points of $C_i$ with cost at least $\left(\frac{z}{\varepsilon}\right)^{2z} \kappa_{C_i}$. More generally for a solution $\mathcal{S}$, let $R_O^{\mathcal{S}}$ denote the union of the outer rings induced by $\mathcal{S}$.

- The main ring $R_M(C_i)$ is the set of points of $C_i$ that are not in the inner or outer rings, i.e., $R_M(C_i) = C_i \setminus (R_I(C_i) \cup R_O(C_i))$.

- For any $j$, the group $G_{j,b}$ consists of the $(2^{b-1}+1)$-th to $(2^b)$-th points of each ring $R_{i,j}$ that arrive in $S$.

- For any $j$, we use $G_{j,\min}$ to denote the union of the groups with the smallest costs, i.e.,
$$G_{j,\min} = \left\{ x | \exists i, x \in R_{i,j}, \mathrm{Cost}(R_{i,j}, \mathcal{A}) < 2 \left(\frac{\varepsilon}{4z}\right)^z \frac{\mathrm{Cost}(R_j, \mathcal{A})}{\beta k} \right\}.$$

- The outer groups $G_b^O$ partition the outer rings $R_O^{\mathcal{A}}$ so that
$$G_b^O = \left\{ x | \exists i, x \in C_i, \left(\frac{\varepsilon}{4z}\right)^z \frac{\mathrm{Cost}(R_O^{\mathcal{A}}, \mathcal{A})}{\beta k} \cdot 2^b \le \mathrm{Cost}(R_O(C_i), \mathcal{A}) < \left(\frac{\varepsilon}{4z}\right)^z \frac{\mathrm{Cost}(R_O^{\mathcal{A}}, \mathcal{A})}{\beta k} \cdot 2^{b+1} \right\}.$$

- We define $G_{\min}^O = \cup_{b \le 0} G_b^O$ and $G_{\max}^O = \cup_{b \ge z \log \frac{4z}{\varepsilon}} G_b^O$.

We remark that unlike the definition of [27], $G_{j,\min}$ is a subset of the groups $G_{j,b}$ with $b \ge 1$, but we shall nevertheless show that our sampling procedure preserves the cost contributed by each group. We also require the following slight variation of the definition of $\mathcal{A}$-approximate centroid set from [53] due to [27].

**Definition B.1** ($\mathcal{A}$-approximate centroid set). *Let $X \subseteq \mathbb{R}^d$ be a set of points, let $k, z$ be two positive integers, and let $\varepsilon > 0$ be an accuracy parameter. Given a set $\mathcal{A}$ of centers, we say a set $\mathbb{C}$ is an $\mathcal{A}$-approximate centroid set for $(k, z)$-clustering on $X$ if for every set of $k$ centers $\mathcal{S} \subseteq \mathbb{R}^d$, there exists $\widetilde{\mathcal{S}} \subseteq \mathbb{R}^d$ of $k$ points such that for all $x \in X$ with $\mathrm{Cost}(x, \mathcal{S}) \le \left(\frac{8z}{\varepsilon}\right)^z \mathrm{Cost}(x, \mathcal{A})$ or $\mathrm{Cost}(x, \widetilde{\mathcal{S}}) \le \left(\frac{8z}{\varepsilon}\right)^z \mathrm{Cost}(x, \mathcal{A})$,*
$$|\mathrm{Cost}(x, \mathcal{S}) - \mathrm{Cost}(x, \widetilde{\mathcal{S}})| \le \frac{\varepsilon}{z \log(z/\varepsilon)} (\mathrm{Cost}(x, \mathcal{S}) - \mathrm{Cost}(x, \mathcal{A})).$$

The following statement is implied by the proof of Theorem 1 in [27].

**Theorem B.2.** *[27, 24] Let $z > 0$ be a constant. Let $x \in G$ for a group induced by an $(\alpha, \beta)$-bicriteria assignment $\mathcal{A}$. For each cluster $C_i$ with $i \in [\beta k]$, let $D_i = C_i \cap G$. Let $\mathbb{C}$ be an $\mathcal{A}$-approximate centroid set for $G$ and let*
$$\gamma = \frac{C \max(\alpha^2, \alpha^z) \beta}{\min(\varepsilon^2, \varepsilon^z)} \log^2 \frac{1}{\varepsilon} \left( k \log |\mathbb{C}| + \log \log \frac{1}{\varepsilon} + \log n \right) \log^2 \frac{1}{\varepsilon},$$
*for some sufficiently large constant $C > 0$. Let*
$$\zeta_x = \frac{\mathrm{Cost}(D_i, \mathcal{A})}{|D_i| \mathrm{Cost}(G, \mathcal{A})} \cdot \gamma \log n, \qquad \eta_x = \frac{\mathrm{Cost}(x, \mathcal{A})}{\mathrm{Cost}(G, \mathcal{A})} \cdot \gamma \log n.$$

*Suppose each point $x \in X$ is sampled and reweighted independently into a set $\Omega_0$ with probability $p_x$, where*
$$p_x \ge \min(\zeta_x + \eta_x, 1).$$
*Let $\Omega_1 = \Omega_0 \setminus (R_I(C_i) \cup (C_i \cap \cup_j G_{j,\min}) \cup (R_O(C_i) \cap G_{\min}^O))$.*

*Suppose $\Omega_2$ is the set of centers in $\mathcal{A}$, where each center $c_i$ with $i \in [\beta k]$ has weight $w_i$, where $w_i$ is a $(1+\varepsilon)$-approximation to $|R_I(C_i)| + |C_i \cap \cup_j G_{j,\min}| + |R_O(C_i) \cap G_{\min}^O|$. Then $(\Omega_1 \setminus \Omega_2) \cup \Omega_2$ is $(1+\varepsilon)$-coreset for the $(k, z)$-clustering problem with probability $1 - \frac{1}{\mathrm{poly}(n)}$.*

We first show that the sampling probabilities for each point in the stream by RINGSAMPLE in Algorithm 1 satisfies the conditions of Theorem B.2.

**Lemma B.3.** *Let $x \in G$ for a group induced by an $(\alpha, \beta)$-bicriteria assignment $\mathcal{A}$ at a time $t$, with $t \in [n]$. For each cluster $C_i$ with $i \in [\beta k]$, let $D_i = C_i \cap G$. Let $\mathbb{C}$ be an $\mathcal{A}$-approximate centroid set for $G$ and let*

$$\gamma = \frac{C \max(\alpha^2, \alpha^z)\beta}{\min(\varepsilon^2, \varepsilon^z)} \log^2 \frac{1}{\varepsilon} \left( k \log |\mathbb{C}| + \log \log \frac{1}{\varepsilon} + \log n \right) \log^2 \frac{1}{\varepsilon},$$

*for some sufficiently large constant $C > 0$ Let*

$$\zeta_x = \frac{\mathrm{Cost}(D_i, \mathcal{A})}{|D_i| \, \mathrm{Cost}(G, \mathcal{A})} \cdot \gamma \log n, \qquad \eta_x = \frac{\mathrm{Cost}(x, \mathcal{A})}{\mathrm{Cost}(G, \mathcal{A})} \cdot \gamma \log n.$$

*Then the probability $p_x$ that RINGSAMPLE (Algorithm 1) samples each point $x$ satisfies*

$$p_x \geq \min(\zeta_x + \eta_x, 1).$$

*Proof.* Suppose that $x \in R_{i,j}$ and $x \in G_{j,b}$ at time $t$, for some $i \in [\beta k]$ in an assignment by $\mathcal{A}$ from MULTMEYERSON. Let $u$ be the time that $x$ arrived in the stream. By the properties of the Meyerson sketch, i.e., MULTMEYERSON in Theorem 2.1, $x$ is irrevocably assigned to a cluster $C_i$ with $i \in [\beta k]$ at time $u$. Hence, $x$ must also be assigned to ring $R_{i,j}$ at time $u$. Moreover, since the stream is insertion-only, then the number of points in all rings $R_{i,j}$ for a fixed $j$ across all $i \in [\beta k]$ is monotonically non-decreasing. Thus $x$ must also be assigned to group $G_{j,b}$ at time $u$.

Let $p_x$ be the sampling probability of $x$ by RINGSAMPLE in Algorithm 1 at time $u$. We have that

$$p_x = \min\left( \frac{4}{r_u} \cdot \gamma \log n, 1 \right),$$

where $r_u$ is the number of points in $G_{j,b}$ at time $u$. Let $G_{j,b}^{(u)}$ be the subset of $G_{j,b}$ that have arrived at time $u$ and let $G_{j,b}^{(t)}$ be the subset of $G_{j,b}$ that have arrived at time $t$. Let $c_i$ be the center assigned to point $x$, so that $\mathrm{Cost}(x, c_i) = \mathrm{Cost}(x, \mathcal{A})$ and let $C_i^{(u)}$ be the points assigned to $c_i$ at time $u$. Similarly, let $D_i^{(u)} = C_i^{(u)} \cap G_{j,b}^{(u)}$. By the definition of $R_{i,j}$ and $G_{j,b}$,

$$\frac{\|x - c_i\|_2^z}{\mathrm{Cost}(G_{j,b}^{(u)}, \mathcal{A})} \leq \frac{2^{j+1}}{\mathrm{Cost}(G_{j,b}^{(u)}, \mathcal{A})} \leq \frac{2^{j+1}}{r_u \cdot 2^j} = \frac{2}{r_u}.$$

Since both the cost of group $G_{j,b}$ and the number of points in $D_i$ is monotonically non-decreasing over time, then at time $t$, we have

$$\frac{\zeta_x}{\gamma \log n} = \frac{\mathrm{Cost}(D_i, \mathcal{A})}{|D_i| \, \mathrm{Cost}(G_{j,b}, \mathcal{A})} \leq \frac{2|D_i|\|x - c_i\|_2^z}{|D_i| \, \mathrm{Cost}(G_{j,b}^{(t)}, \mathcal{A})} \leq \frac{2\|x - c_i\|_2^z}{\mathrm{Cost}(G_{j,b}^{(u)}, \mathcal{A})} \leq \frac{4}{r_u}.$$

Similarly, we have that due to the monotonicity of the cost of group $G_{j,b}$ over time,

$$\frac{\eta_x}{\gamma \log n} = \frac{\|x - c_i\|_2^z}{\mathrm{Cost}(G_{j,b}^{(t)}, \mathcal{A})} \leq \frac{\|x - c_i\|_2^z}{\mathrm{Cost}(G_{j,b}^{(u)}, \mathcal{A})} \leq \frac{2}{r_u}.$$

Thus for sufficiently large constant $C$ in the definition of $\gamma$ in RINGSAMPLE, we have that

$$p_x \geq \min(\zeta_x + \eta_x, 1),$$

since $p_x = \min\left( \frac{4}{r_u} \cdot \gamma \log n, 1 \right)$. $\square$

We next justify the space complexity of Algorithm 1, i.e., showing that with high probability, an upper bound of the number of samples can be determined.

**Lemma B.4.** RINGSAMPLE *(Algorithm 1) samples*

$$O\left( \frac{\max(\alpha^2, \alpha^z)\beta}{\min(\varepsilon^2, \varepsilon^z)} \log^2 \frac{1}{\varepsilon} \left( k \log |\mathbb{C}| + \log \log \frac{1}{\varepsilon} + \log n \right) \log^2 n \log^2 \Delta \log^2 \frac{1}{\varepsilon} \right)$$

*points with high probability.*

*Proof.* Recall that by definition, the groups $G_{j,b}$ partition the points $X = x_1, \ldots, x_n \subseteq [\Delta]^d$. For a fixed $j$ and $b$, let $Y_i$ be an indicator random variable for whether the $i$-th point of $G_{j,b}$ is sampled by RINGSAMPLE. Then we have $\mathbb{E}[Y_i] \leq \frac{4}{i} \cdot \gamma \log n$ and similarly $\mathbb{E}[Y_i^2] \leq \frac{4}{i} \cdot \gamma \log n$. By Bernstein's inequality, Theorem A.4, we have that

$$\mathbf{Pr}\left[\sum Y_i \geq 80\gamma \log^2 n\right] \leq \frac{1}{n^4}$$

and more generally, we have that $\sum Y_i = O\left(\gamma \log^2 n\right)$ with high probability. Thus by a union bound over all $j$ and $b$, we have that the number of sampled points is at most

$$O\left(\gamma \log^2 n \log^2 \Delta\right) = O\left(\frac{1}{\min(\varepsilon^2, \varepsilon^z)} \log^2 \frac{1}{\varepsilon} \left(k \log |\mathbb{C}| + \log \log \frac{1}{\varepsilon} + \log n\right) \log^2 n \log^2 \Delta \log^2 \frac{1}{\varepsilon}\right)$$

for $\gamma = \frac{C \max(\alpha^2, \alpha^z)\beta}{\min(\varepsilon^2, \varepsilon^z)} \log^2 \frac{1}{\varepsilon} \left(k \log |\mathbb{C}| + \log \log \frac{1}{\varepsilon} + \log n\right) \log^2 \frac{1}{\varepsilon}$. $\qquad\square$

Moreover, note that we can for all $t \in [n]$, we can explicitly track both $|G_{j,b}^{(t)}|$ and $\mathrm{Cost}(G_{j,b}^{(t)}, \mathcal{A})$ as the stream is updated, because once the bicriteria algorithm assigns a point to a center in $\mathcal{A}$, the assignment will remain the same for the rest of the stream. Thus, we have the following:

**Lemma B.5.** *For each $j$ and $b$, there exists an algorithm that maintains both $|G_{j,b}^{(t)}|$ and $\mathrm{Cost}(G_{j,b}^{(t)}, \mathcal{A})$ for all $t \in [n]$ using $O\left(\log(nd\Delta)\right)$ space.*

Putting things together, we give the full guarantees of RINGSAMPLE in Algorithm 1.

**Lemma 2.2.** *Let $\mathbb{C}$ be an $\mathcal{A}$-approximate centroid set for a fixed group $G$. There exists an algorithm RINGSAMPLE that samples*

$$O\left(\frac{\max(\alpha^2, \alpha^z)\beta}{\min(\varepsilon^2, \varepsilon^z)} \log^2 \frac{1}{\varepsilon} \left(k \log |\mathbb{C}| + \log \log \frac{1}{\varepsilon} + \log n\right) \log^2 n \log^2 \Delta \log^2 \frac{1}{\varepsilon}\right)$$

*points and with high probability, outputs a $(1 + \varepsilon)$-online coreset for the $k$-means clustering problem.*

*Proof.* Consider RINGSAMPLE. Before claiming the algorithm gives an $(1 + \varepsilon)$-online coreset, we first consider a fixed time $t \in [n]$. Then correctness at time $t$ follows from applying Theorem B.2, given Lemma B.3 and Lemma B.5. We then observe that once a center is formed by RINGSAMPLE, i.e., once a point is sampled, then it irrevocably remains a center in the data structure. Therefore, conditioned on the correctness at time $t$, then the data structure will always correctly give an $(1 + \varepsilon)$-coreset to the prefix of $t$ points in the stream at any later point $t'$ in the stream, $t' \in [n]$ with $t' > t$. It thus suffices to argue correctness over all $t \in [n]$, which requires a simple union bound. The space complexity follows from Lemma B.4 and Lemma B.5. $\qquad\square$

To apply Lemma 2.2, we require upper bounding the term $\log |\mathbb{C}|$. To that end, we first require the following definition of doubling dimension.

**Definition B.6** (Doubling dimension). *The doubling dimension of a metric space $X$ with metric $d$ is the smallest integer $\ell$ such that for any $x \in X$, it is possible to cover the ball of radius $2r$ around $x$ with $2^\ell$ balls of radius $r$.*

Observe that general discrete metric spaces with $n$ points have doubling dimension $O(\log n)$ since all points can be covered by $2^{\log n}$ balls.

We then recall the following result that upper bounds the size $\log |\mathbb{C}|$ for metric spaces with doubling dimension $d$.

**Lemma B.7.** *[27] Given a subset $X$ from a metric space with doubling dimension $d$, $\varepsilon > 0$, and an $\alpha$-approximate solution $\mathcal{A}$ with at most $k \, \mathrm{polylog}(n)$ centers, there exists an $\mathcal{A}$-approximate centroid set for $X$ of size $|X| \cdot \left(\frac{\alpha}{\varepsilon}\right)^{O(d)}$.*

It is known that the Euclidean space has doubling dimension $\Theta(d)$, which would give a $d$ dependency on our coreset size. However, [38] showed that the $d$ dependency can be replaced with $\frac{k}{\varepsilon^2}$, which was subsequently improved by a line of works, e.g., [58, 40, 46], ultimately down to a dependency of $\frac{1}{\varepsilon^2} \log \frac{k}{\varepsilon}$ using the following notion of terminal embeddings:

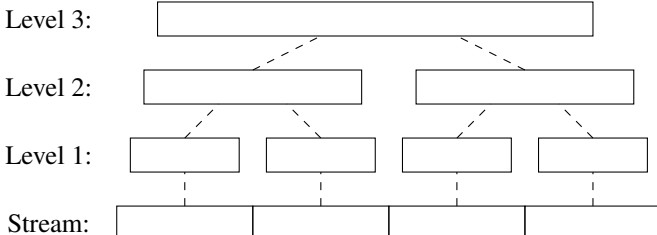

Fig. 2: Merge and reduce framework on a stream of length $n$. The coresets at level 1 are the entire blocks. The coresets at level $i$ for $i > 1$ are each $\left(1 + O\left(\frac{\varepsilon}{2 \log n}\right)\right)$-coresets of the coresets at their children nodes in level $i - 1$.

**Definition B.8** (Terminal embedding). *Let $\varepsilon \in (0,1)$ and $X \subseteq \mathbb{R}^d$ be a set of $n$ points. Then a mapping $f : \mathbb{R}^d \to \mathbb{R}^m$ is a terminal embedding if for all $x \in X$ and all $y \in \mathbb{R}^d$,*

$$(1 - \varepsilon)\|x - y\|_2 \leq \|f(x) - f(y)\|_2 \leq (1 + \varepsilon)\|x - y\|_2.$$

[55] gave a construction of a terminal embedding with $m = O\left(\frac{1}{\varepsilon^2}\log n\right)$ that can be applied in linear space through exhaustive search when polynomial runtime is not required. Thus Lemma 2.2 nows give the following:

**Theorem 1.3.** *There exists an algorithm that samples $\frac{k}{\min(\varepsilon^4, \varepsilon^{2+z})}$ polylog $\frac{n\Delta}{\varepsilon}$ points and with high probability, outputs a $(1 + \varepsilon)$-online coreset for $(k, z)$-clustering.*

For the purpose of clarity, we emphasize that the algorithm does not use sublinear space, even though the sample complexity is sublinear. Namely, for each stream update, we construct and apply a terminal embedding to project each point to a lower dimension. We then compute the appropriate sampling probability of the projected point, but then sample the original point with the computed sampling probability.

## C  Sliding Window Model

In this section, we describe how our online coreset construction for $(k, z)$-clustering on general discrete metric spaces can be used to achieve near-optimal space algorithms for $(k, z)$-clustering in the sliding window model.

We first recall a standard approach for using offline coreset constructions for insertion-only streaming algorithms. Suppose there exists a randomized algorithm that produces an online coreset algorithm that uses $S(n, \varepsilon, \delta)$ points for an input stream of length $n$, accuracy $\varepsilon$, and failure probability $\delta$, where for the ease of discussion, we omit additional dependencies, such as on the dimension $d$, the clustering constraint $k$, the parameter $z$, or additional parameters for whatever problem the coreset construction may approximate. A standard approach for using coresets on insertion-only streams is the merge-and-reduce approach, which partitions the stream into blocks of size $S\left(n, \frac{\varepsilon}{2\log n}, \frac{\delta}{\text{poly}(n)}\right)$ and builds a coreset for each block. Each coreset is then viewed as the leaves of a binary tree with height at most $\log n$, since the binary tree has at most $n$ leaves. Then at each level of the binary tree, for each node in the level, a coreset of size $S\left(n, \frac{\varepsilon}{2\log n}, \frac{\delta}{\text{poly}(n)}\right)$ is built from the coresets representing the two children of the node. Due to the mergeability property of coresets, the coreset at the root of the tree will be a coreset for the entire stream with accuracy $\left(1 + \frac{\varepsilon}{2\log n}\right)^{\log n} \leq (1 + \varepsilon)$ and failure probability $\delta$. We give an illustration of this approach in Figure 2.

This approach fails for sliding window algorithms because the elements at the beginning of the data stream can expire, and so coresets corresponding to earlier blocks of the stream may no longer accurate, which would result in the coreset at the root of the tree also no longer being accurate. On the other hand, suppose we partition the stream into blocks consisting of $S\left(n, \frac{\varepsilon}{2\log n}, \frac{\delta}{\text{poly}(n)}\right)$ elements as before, but instead of creating an offline coreset for each block, we can create an online coreset

for the elements *in reverse*. That is, since the elements in each block are explicitly stored, we can create offline an artificial stream consisting of the elements in the block in reverse and then give the artificial stream as input to the online coreset construction. Note that if we also first consider the "latter" coreset when merging two coresets, then this effectively reverses the stream. Moreover, by the correctness of the online coreset, our data structure provides correctness over any prefix of the reversed stream, or equivalently, any suffix of the stream and specifically, correctness over the sliding window.

Indeed, [9] showed that deterministic online coresets for problems in randomized numerical linear algebra can be used to achieve deterministic algorithms for the corresponding problems in the sliding window model. We thus further adapt the merge-and-reduce framework to show that randomized online coresets for problems in clustering can also be used to achieve randomized algorithms for the corresponding problems in the sliding window model. We formalize this approach in Algorithm 2, duplicated below:

---

**Algorithm 5** Merge-and-reduce framework for randomized algorithms in the sliding window model, using randomized constructions of online coresets

**Input:** A clustering function $f$, a set of points $x_1, \ldots, x_n \subseteq \mathbb{R}^d$, accuracy parameter $\varepsilon > 0$, failure probability $\delta \in (0, 1)$, and window size $W > 0$
**Output:** An approximation of $f$ on the $W$ most recent points
1: Let $\text{CORESET}(X, n, d, k, \varepsilon, \delta)$ be an online coreset construction with $S(n, d, k, \varepsilon, \delta)$ points on a set $X \subseteq \mathbb{R}^d$
2: $m \leftarrow O\left(S\left(n, d, k, \frac{\varepsilon}{\log n}, \frac{\delta}{n}\right) \log n\right)$
3: Initialize blocks $B_0, B_1, \ldots, B_{\log n} \leftarrow \emptyset$
4: **for** each point $x_t$ with $t \in [n]$ **do**
5:     **if** $B_0$ does not contain $m$ points **then**
6:         Prepend $x_t$ to $B_0$, i.e., $B_0 \leftarrow \{x_t\} \cup B_0$
7:     **else**
8:         Let $i$ be the smallest index such that $B_i = \emptyset$
9:         $B_i \leftarrow \text{CORESET}\left(Y, n, d, k, \frac{\varepsilon}{\log n}, \frac{\delta}{n^2}\right)$ for $Y = B_0 \cup \ldots \cup B_{i-1}$   ▷$Y$ is an ordered set of weighted points
10:         **for** $j = 0$ to $j = i - 1$ **do**
11:             $B_j \leftarrow \emptyset$
12:         $B_0 \leftarrow \{x_t\}$
13:     **return** the ordered set $B_{\log n} \cup \ldots \cup B_0$

---

**Theorem 2.3.** *Let $x_1, \ldots, x_n$ be a stream of points in $[\Delta]^d$, $\varepsilon > 0$, and let $X = \{x_{n-W+1}, \ldots, x_n\}$ be the $W$ most recent points. Suppose there exists a randomized algorithm that with probability at least $1 - \delta$, outputs an online coreset algorithm for a $k$-clustering problem with $S(n, d, k, \varepsilon, \delta)$ points. Then there exists a randomized algorithm that with probability at least $1 - \delta$, outputs a coreset for the $k$-clustering problem in the sliding window model with $O\left(S\left(n, d, k, \frac{\varepsilon}{\log n}, \frac{\delta}{n^2}\right) \log n\right)$ points.*

*Proof.* Consider Algorithm 2. Let $\text{CORESET}(X, n, d, k, \varepsilon, \delta)$ be a randomized algorithm that, with probability at least $1 - \delta$, computes an online coreset for a $k$-clustering problem $f$ with $S(n, d, k, \varepsilon, \delta)$ points.

We first claim that for each $B_i$ is a $\left(1 + \frac{\varepsilon}{\log n}\right)^i$ online coreset for $2^{i-1}m$ points. To that end, observe that $B_i$ can only be non-empty if at some time, $B_0$ contains $m$ points and $B_1, \ldots, B_{i-1}$ are all non-empty. By the correctness of the subroutine $\text{CORESET}$, $B_i$ is a $\left(1 + \frac{\varepsilon}{\log n}\right)$ online coreset for the points in $B_0 \cup \ldots \cup B_{i-1}$ at some point during the stream. Hence by induction, $B_i$ is a $\left(1 + \frac{\varepsilon}{\log n}\right)\left(1 + \frac{\varepsilon}{\log n}\right)^{i-1} = \left(1 + \frac{\varepsilon}{\log n}\right)^i$ coreset for $m + \sum_{j=1}^{i-1} 2^{j-1}m = 2^{i-1}m$ points.

Now, because Algorithm 2 inserts the newest points at the beginning of $B_0$, then the stream is fed in reverse to the merge-and-reduce procedure. Thus, for any $W \in [2^{i-1}, 2^i)$, $B_0 \cup \ldots \cup B_i$ provides an online coreset $k$-clustering for the $W$ most recent points in the stream.

To analyze the probability of failure, we remark that there are at most $n$ points in the stream. For each point, there are at most $n$ coresets constructed by the subroutine CORESET (in fact, the number of coreset constructions is upper bounded by $O(\log n)$). Since each subroutine is called with failure probability $\frac{\delta}{n^2}$, then by a union bound, the total failure probability is at most $\delta$.

To analyze the space complexity, note that there are at most $O(\log n)$ coreset constructions $B_0, \ldots, B_{\log n}$ maintained by the algorithm. Each coreset construction samples $S\left(n, d, k, \frac{\varepsilon}{\log n}, \frac{\delta}{n^2}\right)$ points. Hence, the total number of sampled points is $O\left(S\left(n, d, k, \frac{\varepsilon}{\log n}, \frac{\delta}{n^2}\right) \log n\right)$. $\qquad\square$

By Theorem 1.3 and Theorem 2.3, we have:

**Theorem 2.4.** *There exists an algorithm that samples $\frac{k}{\min(\varepsilon^4, \varepsilon^{2+z})}$ polylog $\frac{n\Delta}{\varepsilon}$ points and with high probability, outputs a $(1+\varepsilon)$-coreset to $(k, z)$-clustering in the sliding window model.*

Using an offline algorithm for $(k, z)$-clustering for post-processing after the data stream, we have:

**Theorem 1.1.** *There exists an algorithm that samples $\frac{k}{\min(\varepsilon^4, \varepsilon^{2+z})}$ polylog $\frac{n\Delta}{\varepsilon}$ points and with high probability, outputs a $(1+\varepsilon)$-approximation to $(k, z)$-clustering for the Euclidean distance on $[\Delta]^d$ in the sliding window model.*

# D    Lower Bounds

In this section, we show that any $(1+\varepsilon)$-online coreset for $(k, z)$-clustering requires $\Omega\left(\frac{k}{\varepsilon^2} \log n\right)$ points. The intuition is somewhat straightforward and in a black-box manner. [25] showed the existence of a set $X$ of $\Omega\left(\frac{k}{\varepsilon^2}\right)$ unit vectors such that any sublinear space data structure would not be able to accurately determine $\text{Cost}(C, X)$ for a set of $k$ unit vectors $C$. They thus showed that any offline $(1+\varepsilon)$-coreset construction for $(k, z)$-clustering required $\Omega\left(\frac{k}{\varepsilon^2}\right)$ points.

Because an online $(1+\varepsilon)$-coreset must answer queries on all prefixes of the stream, our goal is to essentially embed $\Omega(\log n)$ instances of the hard instance of [25] into the stream, which would require $\Omega\left(\frac{k}{\varepsilon^2} \log n\right)$ points. To enforce the data structure to sample $\Omega\left(\frac{k}{\varepsilon^2}\right)$ points for each of the hard instance, we give each of the instances increasingly exponential weight. That is, we give the points in the $i$-th instance $\tau^i$ weight for some constant $\tau > 1$, by inserting $\tau^i$ copies of each of the points. Because the weight of the $i$-th instance is substantially greater than the sum of the weights of the previous instances, any $(1+\varepsilon)$-online coreset must essentially be a $(1+\varepsilon)$-coreset for the $i$-th instance, thus requiring $\Omega\left(\frac{k}{\varepsilon^2}\right)$ points for the $i$-th instance. This reasoning extends to all of the $\Omega(\log n)$ instances, thereby giving a lower bound of $\Omega\left(\frac{k}{\varepsilon^2} \log n\right)$ points.

We first recall the following offline coreset lower bound by [25].

**Theorem D.1.** *[25] For $d = \Theta\left(\frac{k}{\varepsilon^2}\right)$, let $X = e_1, \ldots, e_d \in \mathbb{R}^{2d}$ be the set of elementary vectors. Let $z$ be a constant and let $a_1, \ldots, a_m \in \mathbb{R}^{2d}$ with corresponding weights $w_1, \ldots, w_m \in \mathbb{R}$ be a weighted set $P$ of points. Then there exists a set of $k$ unit vectors $C = c_1, \ldots, c_k \in \mathbb{R}^{2d}$ such that for $m = o\left(\frac{k}{\varepsilon^2}\right)$,*

*(1) $\text{Cost}(C, X) = \sum_{i=1}^{d} \min_{j \in [k]} \|e_i - c_j\|_2^2 \geq 2^{z/2} d - 2^{z/2} \cdot \max(1, z/2) \cdot \sqrt{dk}$.*

*(2) $\text{Cost}(C, P) = \sum_{i=1}^{m} w_i \min_{j \in [k]} \|a_i - c_j\|_2^2 < (1 - \varepsilon)(2^{z/2} d - 2^{z/2} \cdot \max(1, z/2) \cdot \sqrt{dk})$.*

We remark that the first property is due to Lemma 31 pf [25] and the second property is due to Lemma 33 and Lemma 34 of [25].

Let $\gamma = \Theta\left(\log \frac{n}{d'}\right)$. Let $d' = \Theta\left(\frac{k}{\varepsilon^2}\right)$ be the dimension of the hard instance in Theorem D.1 and set $d = \gamma d'$, so that we can partition the space $\mathbb{R}^{2d}$ into $\gamma$ groups of $2d'$ coordinates.

We define a stream by creating $\gamma$ weighted instances of the hard instance defined in Theorem D.1. Each of the $\gamma$ hard instances will be embedded into a separate partition of $2d'$ coordinates of $\mathbb{R}^{2d}$. Namely, the first instance consists of the vectors $e_1, \ldots, e_{d'}$ being inserted into the stream. By Theorem D.1, any $(1+\varepsilon)$-coreset must contain $\Omega\left(\frac{k}{\varepsilon^2}\right)$ points. The next instance consists of the

vectors $e_{1+2d'}, \ldots, e_{3d'}$ each being inserted $\tau = 100$ times into the stream. That is, after the vector $e_{d'}$ arrives in the stream from the first hard instance, then $t$ copies of $e_{1+2d'}$ arrive in the stream, followed by then $t$ copies of $e_{2+2d'}$, and so forth. Due to the weights of these vectors, any $(1+\varepsilon)$-coreset must essentially be a $(1+\varepsilon)$-coreset for the second hard instance and thus contain $\Omega\left(\frac{k}{\varepsilon^2}\right)$ points with support in the second group of $2d'$ coordinates.

More generally, for each $i \in [\gamma]$, the stream inserts $t^{i-1}$ copies of $e_{1+2(i-1)d'}$, followed by $\tau^{i-1}$ copies of $e_{2+2(i-1)d'}$, and so on. The main intuition is that due to the weights of the $i$-th group of $d'$ elementary vectors, an $(1+\varepsilon)$-online coreset must contain a $(1+\varepsilon)$-coreset for the $i$-th hard instance. Moreover, since the $(1+\varepsilon)$-online coreset must be a coreset for any prefix of the stream, then it needs to be a $(1+\varepsilon)$-coreset for each of the hard instances. Hence, the online coreset must contain $\gamma \cdot \Omega\left(\frac{k}{\varepsilon^2}\right) = \Omega\left(\frac{k}{\varepsilon^2} \cdot \frac{\log n}{\log \frac{1}{\varepsilon}}\right)$ points.

**Lemma D.2.** *Let $\tau = 100$. For each integer $i > 0$, let $S_i$ be the stream that consists of $\tau^{i-1}$ consecutive copies of $e_{1+2(i-1)d'}$, followed by $\tau^{i-1}$ copies of $e_{2+2(i-1)d'}$, and so on. Let $S$ be the stream that consists of $S_1 \circ S_2 \circ \ldots$. Then for each $i$, any $(1+\varepsilon)$-online coreset after the arrival of $S_i$ must consist of $i \cdot \Omega\left(\frac{k}{\varepsilon^2}\right)$ points.*

*Proof.* We prove the claim by induction on $i$. The base case of $i = 1$ follows from Theorem D.1.

Now suppose the claim holds for a fixed $i-1$. Let $X_i$ be the set of points that have arrived after $S_i$, i.e., $X_i = S_1 \circ \ldots \circ S_i$. Let $C_{i-1}$ be any $(1+\varepsilon)$-online coreset for $S$ after the arrival of $S_{i-1}$. Let $P_i$ be a set of weighted points sampled during stream $S_i$, so that $C_i = C_{i-1} \cup P_i$. Since each point in $S_i$ has weight $\tau^i$, then by scaling the first property of Theorem D.1, we have that there exists a set of $k$ unit vectors $U_i = c_1, \ldots, c_k \in \mathbb{R}^{2d}$ such that

$$
\begin{aligned}
\mathrm{Cost}(U, X_i) &= \sum_{a=1}^{i} \sum_{b=1}^{d'} \tau^a \min_{j \in [k]} \|e_{b+2(a-1)d'} - c_j\|_2^z \\
&\geq \sum_{b=1}^{d'} \tau^i \min_{j \in [k]} \|e_{b+2(i-1)d'} - c_j\|_2^z \\
&\geq (\tau^i)(2^{z/2} d - 2^{z/2} \cdot \max(1, z/2) \cdot \sqrt{dk}).
\end{aligned}
\tag{1}
$$

In particular, the unit vectors $U_i = c_1, \ldots, c_k$ have support entirely in the $i$-th group of $2d'$ coordinates in $\mathbb{R}^{2d}$. By the same argument, there exists a set $U_{i-1}$ with the same properties in the $(i-1)$-th group of $2d'$ coordinates in $\mathbb{R}^{2d}$.

By the correctness of the online coreset, we have

$$
\mathrm{Cost}(U_{i-1}, C_{i-1}) \leq (1+\varepsilon)\,\mathrm{Cost}(U_{i-1}, X_{i-1}) = (1+\varepsilon) \sum_{a=1}^{i-1} \mathrm{Cost}(U_{i-1}, S_a).
$$

Since $U_{i-1}$ consists of unit vectors and each substream $S_a$ consists of unit vectors, then we have

$$
\mathrm{Cost}(U_{i-1}, S_a) \leq 2d' \tau^a.
$$

Thus for $\varepsilon \in (0, 1)$,

$$
\mathrm{Cost}(U_{i-1}, C_{i-1}) \leq 2 \sum_{a=1}^{i-1} (2d' \tau^a) \leq 8d' \tau^{i-1} < \frac{1}{10} d' \tau^i,
$$

since $\tau = 100$. On the other hand, since $U_{i-1}$ has support entirely in the $(i-1)$-th group of $2d'$ coordinates and $S_i$ has support entirely in the $i$-th group of $2d'$ coordinates in $\mathbb{R}^{2d}$, then

$$
\mathrm{Cost}(U_{i-1}, X_i) \geq \mathrm{Cost}(U_{i-1}, S_i) \geq 2d' \tau^i.
$$

Thus for $C_i$ to be a $(1+\varepsilon)$-online coreset for $\varepsilon \in (0, 1)$, $C_i$ must sample additional points from $X_i$ on top of $C_{i-1}$. Hence, $P_i \neq \emptyset$.

In particular, let $P_i$ consist of vectors $y_1, \ldots, y_m$ with weights $w_1, \ldots, w_m$. Since $P_i \neq \emptyset$, then

$$
\mathrm{Cost}(U_i, C_i) = \mathrm{Cost}(U, C_{i-1} \cup P_i) \leq \mathrm{Cost}(U, P_i).
$$

If $|P_i| = o\left(\frac{k}{\varepsilon^2}\right)$, then by the second property of Theorem D.1, we have

$$\text{Cost}(U_i, P_i) = \sum_{b=1}^{m} \min_{j \in [k]} w_b \|y_b - c_j\|_2^2 < \tau^i (1 - \varepsilon)(2^{z/2} d - 2^{z/2} \cdot \max(1, z/2) \cdot \sqrt{dk}),$$

which together with Equation 1 contradicts the fact that $C_i$ is an $(1 + \varepsilon)$-online coreset for $X_i$.

Therefore, we have $|P_i| = \Omega\left(\frac{k}{\varepsilon^2}\right)$. Moreover, since $P_i$ has disjoint support from $C_{i-1}$, then by induction,

$$|C_i| = |C_{i-1} \cup P_i| = |C_{i-1}| + |P_i| = i \cdot \Omega\left(\frac{k}{\varepsilon^2}\right).$$

$\square$

**Theorem 1.4.** *Let $\varepsilon \in (0,1)$. For sufficiently large $n$, $d$, and $\Delta$, there exists a set $X \subset [\Delta]^d$ of $n$ points $x_1, \ldots, x_n$ such that any $(1 + \varepsilon)$-online coreset for $k$-means clustering on $X$ requires $\Omega\left(\frac{k}{\varepsilon^2} \log n\right)$ points.*

*Proof.* Let $\gamma = \Theta\left(\log \frac{n}{d'}\right)$. For each $i \in [\gamma]$, construct the stream $S_i$ as in the statement of Lemma D.2. Observe that $|S_i| = d' \cdot t^i$ for $t = 100$ and so under the settings of the parameter $\gamma$ with the appropriate constant, the total length of the stream $S = S_1 \circ \ldots \circ S_\gamma$ is precisely $n$. Moreover, by Lemma D.2, any $(1 + \varepsilon)$-online coreset must store $\gamma \cdot \Omega\left(\frac{k}{\varepsilon^2}\right) = \Omega\left(\frac{k}{\varepsilon^2} \log n\right)$ points for $n = \text{poly}(d)$. $\square$

# E  On the Proof of Theorem B.2

We remark that Theorem 1 of [27] is stated for sampling a fixed number of points with replacement from each group, rather than sampling each point independently without replacement. By contrast, Theorem B.2 is stated for sampling each point independently without replacement. In this section, we briefly outline the proof of Theorem 1 of [27] and how the analysis translates to the statement of Theorem B.2.

At a high level, the coreset construction of [27] first collects rings of an approximate solution $\mathcal{A}$ of $k$ points into groups, using a similar approach to that described in Appendix B with $\beta = 1$. The algorithm then computes a coreset for each group first using a procedure GROUPSAMPLE and then using a procedure SENSITIVITYSAMPLE for some of the points not considered by the first procedure. We briefly describe both procedures, as well as how to adapt them to the setting where each point is sampled independently and without replacement.

## E.1  Adaptation of Group Sampling

The GROUPSAMPLE procedure of [27] samples a fixed $\Lambda_1$ number of points from each group $G$ with probability proportional to the contribution of each corresponding cluster of the point to the group. That is, given clusters $\widetilde{C}_1, \ldots, \widetilde{C}_k$ induced by $\mathcal{A}$ on $G$, GROUPSAMPLE then performs $\Lambda_1$ rounds of sampling. Each round samples a single point, where a point $p \in \widetilde{C}_i$ is sampled proportional to $\frac{\text{Cost}(\widetilde{C}_i, \mathcal{A})}{|\widetilde{C}_i| \cdot \text{Cost}(G, \mathcal{A})}$ and rescaled appropriately. Then GROUPSAMPLE offers the following guarantees:

**Lemma E.1** (Lemma 2 of [27]). *Let $(X, \text{dist})$ be a metric space, $k$, $z$ be positive integers, $G$ be a group of clients and $\mathcal{A}$ be an $\alpha$-approximate solution to $(k, z)$-clustering on $G$ so that:*

- *For every cluster $\widetilde{C}$ induced by $\mathcal{A}$ on $G$, all points of $\widetilde{C}$ contribute the same cost in $\mathcal{A}$ up to a factor of 2.*

- *For all clusters $\widetilde{C}$ induced by $\mathcal{A}$ on $G$, we have that $\frac{\text{Cost}(G, \mathcal{A})}{2k} \leq \text{Cost}(\widetilde{C}, \mathcal{A})$.*

*Let $\mathbb{C}$ be an $\mathcal{A}$-approximate centroid set for $(k, z)$-clustering on $G$.*

*Then there exists a procedure GROUPSAMPLE that constructs a set $\Omega$ of size*

$$\Lambda_1 = O\left(\frac{\max(\alpha^2, \alpha^z) \log^2 \frac{1}{\varepsilon}}{2^{O(z \log z)} \min(\varepsilon^2, \varepsilon^z)} \left(k \log |\mathbb{C}| + \log \log \frac{1}{\varepsilon} + \log n\right)\right),$$

*such that with high probability, it simultaneously holds for all sets $S$ of $k$ centers that*

$$|\operatorname{Cost}(G, S) - \operatorname{Cost}(\Omega, S)| \leq O\left(\frac{\varepsilon}{\alpha}\right)(\operatorname{Cost}(G, S) + \operatorname{Cost}(G, \mathcal{A}).$$

We outline the high-level approach of the proof of Lemma E.1 and how can it can adjusted for an $(\alpha, \beta)$-approximate solution $\mathcal{A}$, as well as a process that samples each point independently without replacement, rather than using $\Lambda_1$ rounds as GROUPSAMPLE.

The proof of Lemma E.1 involves further partitioning the points of $G$ into three subsets, based on the cost induced by the point. Namely, given a set $S$ of $k$ centers, a point $p$ in group $G$ is categorized as tiny, interesting, or huge, depending on $\operatorname{Cost}(p, S)$ (though the interesting and huge points actually have a small overlap to allow slack in the proof). [27] applies standard Chernoff bounds to show that the number of sampled points is well-concentrated around its expectation and then applies Bernstein's inequality to show that the clustering costs of the tiny points, the interesting points are well-concentrated around their expectations. In particular, they show that the expected number of sampled points from each cluster $\widetilde{C_i}$ is

$$\frac{\Lambda_1 \operatorname{Cost}(\widetilde{C_i}, \mathcal{A})}{\operatorname{Cost}(G, \mathcal{A})} \geq \frac{\Lambda_1}{2k},$$

due to the assumption that for all clusters $\widetilde{C}$ induced by $\mathcal{A}$ on $G$, we have that $\frac{\operatorname{Cost}(G, \mathcal{A})}{2k} \leq \operatorname{Cost}(\widetilde{C}, \mathcal{A})$.

We first remark that if $\mathcal{A}$ is an $(\alpha, \beta)$-approximate solution rather than an $\alpha$-approximate solution, i.e., if $\mathcal{A}$ has $\beta k$ centers rather than $k$ centers, then the definition of the rings and groups would instead insist that for all clusters $\widetilde{C}$ induced by $\mathcal{A}$ on $G$, we have that $\frac{\operatorname{Cost}(G, \mathcal{A})}{2\beta k} \leq \operatorname{Cost}(\widetilde{C}, \mathcal{A})$. Then by oversampling $\Lambda_1$ by a factor of $\beta$, i.e., sampling $\beta\Lambda_1$ points would ensure that the expected number of sampled points from each cluster $\widetilde{C_i}$ would be

$$\frac{\beta\Lambda_1 \operatorname{Cost}(\widetilde{C_i}, \mathcal{A})}{\operatorname{Cost}(G, \mathcal{A})} \geq \frac{\beta\Lambda_1}{2\beta k} = \frac{\Lambda_1}{k}.$$

It then remains to argue the correctness of sampling each point independently without replacement rather than a fixed $\beta\Lambda_1$ number of points, which simply holds by adjusting the applications of the Chernoff bounds and Bernstein's inequality so that there is a separate random variable for each point in the input rather than for each of the $\Lambda_1$ rounds.

### E.2 Adaptation of Sensitivity Sampling

The SENSITIVITYSAMPLE procedure of [27] samples a fixed $\Lambda_2$ number of points from each group $G$ with probability proportional to the contribution of the point. Specifically, SENSITIVITYSAMPLE then performs $\Lambda_2$ rounds of sampling, where each round samples a point $p$ in the group $G$ with probability proportional to $\frac{\operatorname{Cost}(p, \mathcal{A})}{\operatorname{Cost}(G, \mathcal{A})}$ and rescales the sampled point appropriately. Then SENSITIVITYSAMPLE offers the following guarantee:

**Lemma E.2** (Lemma 3 of [27])**.** *Let $(X, \operatorname{dist})$ be a metric space, $k$, $z$ be positive integers, and $\mathcal{A}$ be an $\alpha$-approximate solution to $(k, z)$-clustering on $G$. Let $\mathbb{C}$ be an $\mathcal{A}$-approximate centroid set for $(k, z)$-clustering on $G$. Let $G$ be either a group $G_b^O$ or $G_{\max}^O$. Then there exists a procedure* SENSITIVITYSAMPLE *that constructs a set $\Omega$ of size*

$$\Lambda_2 = O\left(\frac{2^{O(z \log z)} \alpha^2 \log^2 \frac{1}{\varepsilon}}{\varepsilon^2}\left(k \log |\mathbb{C}| + \log \log \frac{1}{\varepsilon} + \log n\right)\right),$$

*such that with high probability, it simultaneously holds for all sets $S$ of $k$ centers that*

$$|\operatorname{Cost}(G, S) - \operatorname{Cost}(\Omega, S)| \leq O\left(\frac{\varepsilon}{\alpha z \log \frac{z}{\varepsilon}}\right)(\operatorname{Cost}(G, S) + \operatorname{Cost}(G, \mathcal{A}).$$

We outline the high-level approach of the proof of Lemma E.2 and how can it can adjusted for an $(\alpha, \beta)$-approximate solution $\mathcal{A}$, as well as a process that samples each point independently without replacement, rather than using $\Lambda_2$ rounds as SENSITIVITYSAMPLE.

The proof of Lemma E.2 partitions the points of $G$ into two categories, based on the cost induced by the point. Given a set $S$ of $k$ centers, the close points are the points $p$ in $G$ that have $\text{Cost}(p, S) \leq 4^z \cdot \text{Cost}(p, \mathcal{A})$. The far points are the remaining points in $G$, i.e., the points $p$ in $G$ with $\text{Cost}(p, S) > 4^z \cdot \text{Cost}(p, \mathcal{A})$.

[27] applies Bernstein's inequality to show that the clustering cost of the close points is well-concentrated around their expectations. We can again adjust the application of Bernstein's inequality so that there is a separate random variable for each point in the input rather than for each of the $\Lambda_2$ samples.

To handle the far points, [27] again uses Bernstein's inequality to show that with high probability, the clustering points of these points with respect to $S$ can be replaced with the distance to the closest center $c \in \mathcal{A}$ plus the distance from $c$ to the closest center in $S$. Conditioned on this event, the latter distance can then be charged to the remaining points of the cluster from the original dataset, i.e., the remaining points of the cluster not necessarily restricted to group $G$, which are significantly more numerous and already paying a similar value in $S$. In particular, Bernstein's inequality utilizes the fact that the second moment of the estimated cost of a cluster $C$ is at most

$$\frac{\text{Cost}(G, \mathcal{A})}{\Lambda_2^2} \text{Cost}(C \cap G, \mathcal{A}) \leq \frac{2k}{\Lambda_2^2} (\text{Cost}(C \cap G, \mathcal{A}))^2,$$

for $\beta = 1$. Thus for general $\beta$, we recover the same guarantee by oversampling $\Lambda_2$ by a factor of $\beta$, i.e., sampling $\beta \Lambda_2$ points would ensure that the second moment would be at most $\frac{2k}{\Lambda_2^2} \text{Cost}^2(C \cap G, \mathcal{A})$. It then remains to argue the correctness of sampling each point independently without replacement rather than a fixed $\beta \Lambda_2$ number of points, which again holds by adjusting the application of Bernstein's inequality so that there is a separate random variable for each point in the input rather than for each of the $\Lambda_2$ rounds.

## F    Additional Experiments on Synthetic Data

We first describe the methodology and experimental setup of our empirical evaluation on a synthetic dataset before detailing the experimental results. To emphasize the benefits of our algorithm against worst-case input, we generate a synthetic dataset that would fully capture the failure cases of previous baselines.

**Dataset.**    We generated our dataset $X$ consisting of $200,001$ points on two-dimensional space so that $100,000$ points were drawn from a spherical Gaussian with standard deviation $2.75$ centered at $(-10, 10)$ and $100,000$ points were drawn from a spherical Gaussian with standard deviation $2.75$ centered at $(10, -10)$. The final point of $X$ was drawn from a spherical Gaussian with standard deviation $2.75$ centered at $(100000, 100000)$. Thus by construction of our synthetic dataset for $k = 3$, the optimal centers should be close to $(-10, 10)$, $(10, -10)$, and $(100000, 100000)$. We then create the data stream $S$ by prepending two additional points drawn from spherical Gaussians with standard deviation $2.75$ centered at $(-100000, 100000)$ and $(-100000, -100000)$ respectively. We set the window length to be $200,001$ in accordance with the "true" data set, so that the first two points of the stream of length $200,003$ will be expired by the data stream.

**Experimental setup.**    For each of the instances of Lloyd's algorithm, either on the entire dataset $X$ or the sampled coreset $C$, we use 3 iterations using the k-means++ initialization. In this case, the offline Lloyd's algorithm requires storing the entire dataset $X$ in memory and thus its input size is $200,001$ points. By comparison, we normalize the space requirement of the sublinear-space algorithms by permitting each algorithm to store $m \in \{3, 4, 5, 6, 7, 8, 9, 10, 11, 12\}$ points. Note that since $k = 3$, it would not be reasonable for $C$ to have fewer than 3 points. We then run Lloyd's algorithm on the coreset $C$, with 3 iterations using the k-means++ initialization.

By construction of our dataset, we generally expect the uniform sampling algorithm uni to be stable across the various values of $m$ but perform somewhat poorly, as it will sample points from the large clusters but it will miss the point generated from the Gaussian centered at $(100000, 100000)$. Since in our construction the stream $S$ only contains two more points than the dataset $X$, the histogram-based algorithm hist will not delete any points. Thus, the resulting coreset $C$ generated by hist is somewhat likely contain the points generated from the Gaussians centered at $(-100000, 100000)$

and $(-100000, -100000)$ and can perform poorly on the synthetic dataset in these cases. Finally, since we allow the last point of the stream to be the single point of $X$ far from the two large clusters, then the importance sampling based algorithm imp will sample the last point with high probability once any points of $C$ have been expired. Hence by the construction of our stream, we expect imp to perform well.

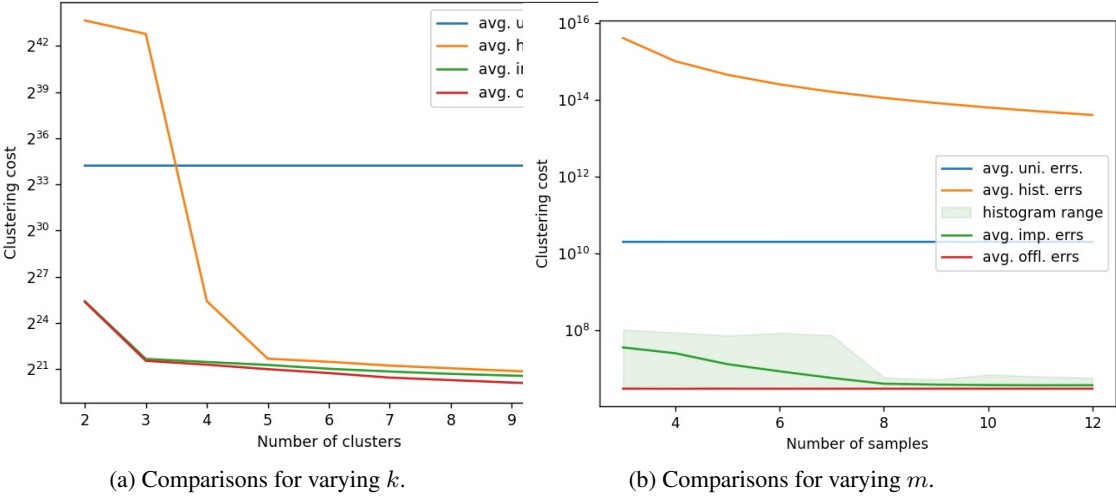

(a) Comparisons for varying $k$.  (b) Comparisons for varying $m$.

Fig. 3: Comparison of average clustering costs made by uniform sampling, histogram-based algorithm, and our coreset-based algorithm across various settings of space allocated to the algorithm, given a synthetic dataset. For comparison, we also include the offline k-means++ algorithm as a baseline, though it is inefficient because it stores the entire dataset. Ranges are not plotted because they would not be visible.

**Experimental results.** For each choice of $m$ and $k$, we ran each algorithm 50 times and tracked the resulting clustering cost. As expected by our construction, our algorithm performed significantly better than the other sublinear-space algorithms. In fact, even though our algorithm was only permitted memory size $m \in \{3, 4, 5, 6, 7, 8, 9, 10, 11, 12\}$, our algorithm was quite competitive with the offline Lloyd's algorithm, which used memory size $200, 001$, i.e., the entire dataset. For $k \geq 3$, uniform sampling performed relatively poorly but quite stably, because although it never managed to sample the point generated from the Gaussian centered at $(100000, 100000)$, the two other Gaussian distributions were sufficiently close that any sampled point would serve as a relatively good center for points generated from the two distributions. Similarly, for fixed $k = 3$ in Figure 3b, the importance sampling approach used by histogram-based algorithms performed the worse, by multiple orders of magnitude. We expect this is because we did not delete the points in $S \setminus X$ from $C$ and thus the resulting Lloyd's algorithm on $C$ moved the centers far away from the centers of the Gaussian distributions that induced $X$. A more optimized fine-tuned histogram-based algorithm would have searched for parameters that govern when to delete points from $S \setminus X$, which have reduced the algorithm down to our main algorithm. We plot our results in Figure 3.

