# OpenReview forum: "Near-Optimal $k$-Clustering in the Sliding Window Model"
_NeurIPS.cc/2023/Conference — NeurIPS 2023 poster_

### Official Review · Reviewer_o3G5 · 2023-06-13

**Soundness:** 3 good
**Presentation:** 3 good
**Contribution:** 3 good
**Rating:** 7
**Confidence:** 4

**Summary:**

This paper proposes the first near-optimal $(1+\varepsilon)$-approximation algorithm for $(k, z)$-clustering problem in the sliding window model. The core part is the $(1+\varepsilon)$-coreset of $(k, z)$-clustering in the sliding window model which is based on an online coreset algorithm for $k$-clustering problem. This paper gives strict proof for the proposed algorithm and experimental results manifest its efficiency. In short, the theory aspect of this paper is solid enough.

**Strengths:**

(1) This paper gives the first $(1+\varepsilon)$-approximation algorithm for $(k, z)$-clustering in the sliding window, which improves the existing results in two aspects including accuracy and space.
(2) The space $\frac{k}{\min (\varepsilon^4, \varepsilon^{2+z}) }$ of $(1+\varepsilon)$-coreset for $(k, z)$-clustering in the sliding window almost matches the lower bound $\Omega \left(\frac{k}{\varepsilon^2} \log{n}\right)$.
(3) Strict proof is provided and experimental results are sufficient.
(4) The paper is well-written and well-organized.

**Weaknesses:**

(1) Some symbol is not explained, like $[ \Delta ]^d$.
(2) The merge-and-reduce framework lacks some citations.


**Questions:**

In Line 92-93, Theorem 1.1 presents the word space $\frac{k}{\min(\varepsilon^4, \varepsilon^{2+z})} \text{poly} \log \frac{n \Delta}{\varepsilon}$. For $k$-median and $k$-means, $z$ is equal to $1$ and $2$ respectively. No matter $z$ is $1$ or $2$, the space should be $\frac{k}{\varepsilon^4} \text{poly} \log \frac{n \Delta}{\varepsilon}$ since $\varepsilon \in (0, 1)$. How can you achieve $\frac{k}{\varepsilon^2} \text{poly} \log \frac{n \Delta}{\varepsilon}$? In addition, for $(k, z)$-clustering, $z=1$ does not correspond to $k$-median when $\mathrm{dist}$ represents the Euclidean distance, since $k$-median is $\ell_1$-norm.

**Limitations:**

See Weaknesses and Questions.

---

> ### Author Rebuttal · Authors · 2023-08-09
>
> > Some symbol is not explained, like $[\Delta]^d$.
>
> We will clarify that $[\Delta]^d$ means that each of the $d$ coordinates of each point must lie within $\{1,\ldots,\Delta\}$.
>
> > The merge-and-reduce framework lacks some citations.
>
> We will add references to previous work that use the merge-and-reduce framework.
>
> > No matter $z$ is $1$ or $2$, the space should be $\frac{k}{\varepsilon^4}\text{poly} \log\frac{n\Delta}{\varepsilon}$
> since $\varepsilon\in(0,1)$. How can you achieve $\frac{k}{\varepsilon^2}\text{poly} \log\frac{n\Delta}{\varepsilon}$?
>
> We emphasize that our results do not claim to achieve $\frac{k}{\varepsilon^2}\text{poly} \log\frac{n\Delta}{\varepsilon}$ words of space but rather $\frac{k}{\varepsilon^{z+2}}\text{poly} \log\frac{n\Delta}{\varepsilon}$ words of space. Moreover, in light of an $\Omega\left(\frac{k}{\varepsilon{2+z}}\log n\right)$ lower bound by [41], it is not possible to achieve $\frac{k}{\varepsilon^2}\text{poly} \log\frac{n\Delta}{\varepsilon}$.
>
> > In addition, for $(k,z)$-clustering, $z=1$ does not correspond to $k$-median when $\text{dist}$ represents the Euclidean distance, since $k$-median is $\ell_1$-norm.
>
> Since $(k,z)$-clustering is a general definition that is the sum of the $z$-th power of the distances, then $k$-median with the Euclidean distance is the sum of the Euclidean distances. Nevertheless, $k$-median is also well-defined when the underlying distance is the Manhattan distance, though this setting is beyond the current scope of our paper.

---

> > ### Comment · Reviewer_o3G5 · 2023-08-14
> >
> > I don't think you answer my questions directly.
> > (1) In Theorem 1.1, the words of space is $\frac{k}{\min(\epsilon^4, \epsilon^{2+z})} \text{polylog}\frac{n \Delta}{\epsilon}$. In Line 92-93, no matter $z=1$ or $z=2$, it would be $\frac{k}{\epsilon^4} \text{polylog}\frac{n \Delta}{\epsilon}$ for $\epsilon \in (0, 1)$, right? However, this paper claims that it achieves $\frac{k}{\epsilon^2} \text{polylog}\frac{n \Delta}{\epsilon}$ words of space!
> > (2) In your definition of $(k, z)$-clustering, its the summation of $\ell_2$-norm, but $k$-median is the summation of $\ell_1$-norm. Please keep strict.

---

> > > ### Author Response · Authors · 2023-08-15
> > >
> > > Thanks for the follow-up! We provide specific responses to your points below -- please let us know if you have any further questions.
> > >
> > > 1) Ah yes, thanks for catching the typo! Line 92-93 should claim a space bound of $\frac{k}{\varepsilon^4}\text{polylog}\frac{n\Delta}{\varepsilon}$ rather than $\frac{k}{\varepsilon^2}\text{polylog}\frac{n\Delta}{\varepsilon}$. Note that this is consistent with Theorem 1.1 for $z=1$ and $z=2$ (as well as more general values of $z$), since $\min(\varepsilon^4,\varepsilon^{2+z})=\varepsilon^4$ for $z\le 2$ as you pointed out. Moreover, this is consistent with Table 1, the claimed bounds in the abstract, and the lower bound of $\frac{k}{\varepsilon^{2+z}}$ by [41].
> > >
> > > 2) We would like to emphasize that $(k,z)$-clustering is defined as the sum of the $z$-th power of the distances in some fixed metric, e.g., see [23, 24, 41, 42, 47]. Thus when the fixed metric is the Euclidean distance, then $k$-median, which corresponds to $z=1$, is the sum of the Euclidean distances, i.e., the sum of the $\ell_2$ distances, NOT the sum of the $\ell_1$ distances. For example, note that [42] furthermore explicitly defines $(k,z)$-clustering as the sum of the $z$-th power of the *Euclidean* distances. We also remark that it is possible to study this problem for all ranges of $z$ when the underlying metric is the Manhattan instance, i.e., the $\ell_1$ norm. However, this setting is beyond the current scope of our paper.

---

> > > > ### Comment · Reviewer_o3G5 · 2023-08-15
> > > >
> > > > Thank you for your responses!

---

### Official Review · Reviewer_YLu7 · 2023-07-04

**Soundness:** 3 good
**Presentation:** 4 excellent
**Contribution:** 3 good
**Rating:** 7
**Confidence:** 2

**Summary:**

This paper studies the $k$-clustering problem in sliding window model. In sliding window model, a window of size $W$ is given to capture the most recent $W$ updates in the stream, where good clustering approximation should be maintained in the window with small space complexity. In previous work, in order to achieve an $(1+\epsilon)$ approximation for $(k,z)$-clustering problem in sliding window, a space complexity of $\frac{kd+d^{Cz}}{\epsilon^3}polylog(W,\Delta,\frac{1}{\epsilon})$ for some constant $C \ge 7$. In this work, the authors propose a coreset-based method, which gives an $(1+\epsilon)$-approximation for the $(k,z)$-clustering problem in sliding window model with space complexity $\frac{k}{min(\epsilon^4,\epsilon^{2+z})}polylog(\frac{n\Delta}{\epsilon})$, which is independent of the window size and nearly matches the lower bound of the space used by the offline coreset construction.

**Strengths:**

1. The presented result significantly improves the space complexity of previous work with $(1+\epsilon)$-approximation guarantee on the clustering quality in sliding window model. Besides, this is the first approximation result that can achieve a space complexity independent of the window size $W$. In practical settings, since $\Delta$ is usually assumed to be bounded by $poly(n)$, the presented result can nearly match the lower bound space complexity required for any $(1+\epsilon)$-online coreset construction for $(k,z)$-clustering problem.

2. The extensive experimental results on real world datasets show that the proposed method is more efficient than previous ones.

**Weaknesses:**

1. The techniques used in this paper seem to rely heavily on the current SOTA method for offline coreset construction [1] (using ring structures and independent sampling method) and the consistent approximation scheme proposed by Meyerson et al. [2].

2. The challenges for merge and reduce operations is not discussed comprehensively in the literature.

[1] Adam Meyerson. Online facility location. In 42nd Annual Symposium on Foundations of Computer Science, FOCS, pages 426–431. IEEE Computer Society, 2001.

[2] Vincent Cohen-Addad, Kasper Green Larsen, David Saulpic, and Chris Schwiegelshohn. Towards optimal lower bounds for k-median and k-means coresets. In STOC ’22: 54th Annual ACM SIGACT Symposium on Theory of Computing, pages 1038–1051, 2022.

**Questions:**

1. It seems that the $O(log\Delta)$ dependence on the space complexity comes from the Meyerson approach (for obtaining the consistent approximation), where a guess-and-double process is used to obtain an estimation of the cost of an optimal solution. Is the $O(log\Delta)$ term necessary in the proposed method and is there any other method that can avoid guessing $O(log\Delta)$ times for the optimal clustering cost (for example by calling a single-criteria approximation algorithm with large approximation ratio to serve as the upper bound for optimal clustering cost)?

2. Can the authors explain more about the challenges when directly applying the reduce and merge method for each block of the divided data points in a stream without using an "inverse" operation?

3. I am curious about whether, in the sliding window model, the space complexity is more critical to consider compared to the time complexity. As I am not very familiar with this model, I would appreciate some insights on this matter.

**Limitations:**

Since this is mainly a theoretical work, I don't think there is potential negative societal impact of this work.

---

> ### Author Rebuttal · Authors · 2023-08-09
>
> > The techniques used in this paper seem to rely heavily on the current SOTA method for offline coreset construction [1] (using ring structures and independent sampling method) and the consistent approximation scheme proposed by Meyerson et al. [2].
>
> Our main algorithm utilizes (1) a framework for sliding window algorithms using a new randomized online coreset construction along the lines of [8], (2) an online version of the coreset construction of [24], and (3) a consistent assignment across all times of the stream through an online facility location argument in the style of [30]. Although none of these steps are particularly complicated, we believe that developing the correct variations of each step and putting them together in the correct manner is fairly novel/technical, e.g., the long line of previous work for clustering on sliding windows has missed this approach.
>
> > It seems that the $O(log\Delta)$ dependence on the space complexity comes from the Meyerson approach (for obtaining the consistent approximation), where a guess-and-double process is used to obtain an estimation of the cost of an optimal solution. Is the $O(log\Delta)$ term necessary in the proposed method and is there any other method that can avoid guessing $O(log\Delta)$ times for the optimal clustering cost (for example by calling a single-criteria approximation algorithm with large approximation ratio to serve as the upper bound for optimal clustering cost)?
>
> While our lower bound shows that an $\Omega(\log n)$ dependence is necessary for any online coreset, it is indeed unclear whether an $O(\log\Delta)$ dependence is necessary. We leave this as a great question for future work.
>
> > The challenges for merge and reduce operations is not discussed comprehensively in the literature...Can the authors explain more about the challenges when directly applying the reduce and merge method for each block of the divided data points in a stream without using an "inverse" operation?
>
> The main barrier is that existing merge-and-reduce frameworks using coresets do not handle the implicit deletions of the sliding window model. Thus we instead show that the existing coreset construction of [1] can be modified in an online manner, and then show that there exists a corresponding merge-and-reduce framework for online coresets.
>
> > I am curious about whether, in the sliding window model, the space complexity is more critical to consider compared to the time complexity. As I am not very familiar with this model, I would appreciate some insights on this matter.
>
> Yes, in the sliding window model, including the more specific streaming model, the main quantity of interest in previous work is the space complexity, though considerations for time complexity should be optimized as much as possible. Towards that end, we remark that the calculations of the sampling probailities of our algorithm only require computing the distance of each point to a number of centers, as well as tracking the number of points in each ring/group so far. Thus, our algorithm can be efficiently implemented, and has a small polynomial running time.

---

> > ### Comment · Reviewer_YLu7 · 2023-08-19
> >
> > I would like to thank the authors for their response and clarification. I think this is an interesting paper.

---

### Official Review · Reviewer_P4Gj · 2023-07-08

**Soundness:** 3 good
**Presentation:** 3 good
**Contribution:** 3 good
**Rating:** 7
**Confidence:** 4

**Summary:**

This paper studies $k$-means and $k$-median clustering in the sliding window, and proposes an $(1+\varepsilon$)-approximation algorithm on the top of an coreset maintained through the stream. The space complexity is roughly $k/\text{poly}(\varepsilon) \cdot \text{poly}\log n$ where $n$ is the number of points, coming together with an nearly-tight lower bound.

**Strengths:**

- The paper is well-written in general, most parts are clear and well-organized.
- The results are interesting and significant. I did not read the entire proof, but checking the key steps convince me the result should be correct. I am happy to go back to proof details if any concern comes up.
- It is a plus for a theory paper to have experiments. The space consumption is reduced a lot with some compromise on the clustering performance.

**Weaknesses:**

- I am confused by line 388 389, it seems that the window size is too close to the stream size. Though theoretically the window size is not an important parameter, but doing this converges back to the streaming model.
- After staring at it for a while, I do not think either of the algorithm or lower bound applies to the $k$-center clustering. Then is it OK to have $k$-clustering as in the title? The authors can either convince me this result holds / has impact on $k$-center clustering, or show me a bunch of work that does not include $k$-center in $k$-clustering.
- I think it is good to make clear of the notation, especially when it first appears. In the abstract, maybe clarify what $z, \Delta$ is.

**Questions:**

- Will the codes be made public?

- Can you add an open problem section?

**Limitations:**

I would appreciate this as a basically-complete theory work. There are some limitations on the experiments, but not a hurdle on assessing the contribution of this paper.

---

> ### Author Rebuttal · Authors · 2023-08-09
>
> > I am confused by line 388 389, it seems that the window size is too close to the stream size. Though theoretically the window size is not an important parameter, but doing this converges back to the streaming model.
>
> We remark that the window size was set rather close to the stream size in our experiments as a simple proof-of-concept where sampling-based algorithms demonstrate better performance than histogram-based algorithms. We believe that more general datasets with different settings for window size and stream size can also be constructed so that sampling-based algorithms demonstrate better performance than histogram-based algorithms.
>
> > After staring at it for a while, I do not think either of the algorithm or lower bound applies to the $k$-center clustering. Then is it OK to have $k$-clustering as in the title? The authors can either convince me this result holds / has impact on $k$-center clustering, or show me a bunch of work that does not include $k$-center in $k$-clustering
>
> Examples of works that do not include $k$-center in $k$-clustering are listed below. Nevertheless for the sake of clarity, we can change the title to "Near-Optimal Clustering in the Sliding Window Model" if permitted.
>
> Artur Czumaj, Guichen Gao, Shaofeng H.C. Jiang, Robert Krauthgamer, Pavel Vesely: Fully Scalable MPC Algorithms for Clustering in High Dimension. CoRR abs/2307.07848 (2023)
>
> Yecheng Xue, Xiaoyu Chen, Tongyang Li, Shaofeng H.-C. Jiang: Near-Optimal Quantum Coreset Construction Algorithms for Clustering. CoRR abs/2306.02826 (2023)
>
> Sayan Bandyapadhyay, Fedor V. Fomin, Tanmay Inamdar: Coresets for Clustering in Geometric Intersection Graphs. SoCG 2023: 10:1-10:16
>
> Lingxiao Huang, Shaofeng H.-C. Jiang, Jianing Lou, Xuan Wu: Near-optimal Coresets for Robust Clustering. ICLR 2023
>
> Vincent Cohen-Addad, Alessandro Epasto, Vahab Mirrokni, Shyam Narayanan, Peilin Zhong: Near-Optimal Private and Scalable $k$-Clustering. NeurIPS 2022
>
> > I think it is good to make clear of the notation, especially when it first appears. In the abstract, maybe clarify what $z,\Delta$ is.
>
> Thanks for the suggestion. We will clarify in the abstract that $z$ is the power of the distance in the cost function and $\Delta$ is the length of the grid on which the points lie.
>
> > Will the codes be made public?
>
> Yes, we will upload the code to a public repository (and hopefully it should already be accessible in the supplementary material).
>
> > Can you add an open problem section?
>
> Yes, we will add a conclusion that both summarizes our contributions, as well as provide directions for future work.
>
> > I would appreciate this as a basically-complete theory work. There are some limitations on the experiments, but not a hurdle on assessing the contribution of this paper.
>
> Thanks for the feedback! We can move the experiments to the appendix in the full version so that the main body focuses on the theory aspects of the paper.

---

> > ### Comment · Reviewer_P4Gj · 2023-08-14
> >
> > Thank the authors for the response. At this moment, I am happy with most issues. But on the window size in the experiments, I still think having a generic window size $W$ (actually try with different $W$) is important to demonstrate the performance remains immune to $W$, which corroborates with the result that $W$ does not contribute to the sample complexity.  This does not violate the motivation of showing sampling-based algorithms perform better than histogram-based algorithms.
> >
> > More importantly, we should be aware of a recent result (https://research.google/pubs/pub52480) breaking the barrier of $\log(n\Delta)$ memory of words. While the result of this paper lies in $O(\text{polylog}(n\Delta))$. It is a bit pity that the full version is not yet available, it would be interesting to follow up if their technique can be used for sliding window model.
> >
> > I will maintain my evaluation on acceptance.

---

### Official Review · Reviewer_8X17 · 2023-07-14

**Soundness:** 4 excellent
**Presentation:** 2 fair
**Contribution:** 3 good
**Rating:** 7
**Confidence:** 4

**Summary:**

The paper proposes an near-optimal algorithm to build (1+$\epsilon$)-online coreset for k-clustering problem, and apply it to the sliding window model. The (near) optmality is established by a matching lower bound, also proved in the paper.

A coreset is a compression of a dataset, such that any clustering centers will induce similar cost over the coreset and the original dataset. The (insertion-only) online coreset problem aims to build a coreset incrementally for a data stream, such that the coreset property (i.e., cost approximation) holds at any time point

The main algorithm borrows lots of ideas from paper [24] ("*A New Coreset Framework for Clustering*"), which proposes a framework for building (offline) coresets and can be roughly described as follows:
1. Given a dataset, find a constant-approx clustering solution $A$ on it.
2. For each cluster induced by $A$, divide it into rings by distance to the center, such that all points in a same ring have roughly the same distance to the center.
3. Group rings at the same level from each cluster, then do importance sampling on the group. The sampled points (with scaled weights) will be the output coreset.
For analysis, [24] shows that: as long as you can find a small "$A$-approximate centroid" $C$ (which is essentially something like a $\epsilon$-net restricted to part of the dataset), then you can have a small coreset.

The paper essentially adapts the algorithm from [24] to the online setting
1. To get an constant-approx clustering in the online setting, the paper adopts a bi-criteria variant of Meyerson's algorithm for online facility location. Here the algorithm could produce more than k centers, but this is not an issue for bounding the coreset size since it only adds a constant factor.
2. The importance sampling step is mostly the same as described before, just in an online fashion. This gives the online coreset construction.
3. Finally, to apply it in the sliding window model, the paper breaks the input stream into blocks, then apply the online coreset construction in the reverse order on each block, and merge each block's coreset also in the reverse order (and keep only those within the window).

I don't have time to look into the details of the lowerbound construction, however, it appears to be an adaptation of the offline bound from [23] ("*Towards optimal lower bounds for k-median and k-means coresets*").


**Strengths:**

I think the main contribution is mostly in the theory side. The problem studied is important and well-motivated. It's nice to see that the result of [23] and [24] can be ported to the online setting. This requires quite some non-trivial work and the paper is able to identify an optimal choice of parameters, which results in an asymptotically optimal algorithm.

**Weaknesses:**

As mentioned in the Summary, the main idea seems to be an (arguably) straightforward adaptation of [24], which somewhat limits the novelty.

Also, I feel the experiment section is kinda redundant or even harmful to the paper's contribution. The proposed algorithm is not acutally implemented (in particular, the RingSample Alg, which is a core part of the proposed algorithm). It seems that the performance gain comes only from discarding points outside the sliding window, especially when compared with the histogram-based algorithm.
- If the authors are able to implement their algorithm in a more meaningful way, and if it indeed produces better result on some realworld dataset, then I would be happy to increase my rating. Otherwise, I feel it's better to just remove the experiment section completely and present the paper as a theoretical contribution.

Lastly, I believe the writing can be improved a lot. It seems the draft may have been finalized hastily, with instances of noticeable cutting and pasting to meet the page limit. This renders the main body insufficiently self-contained and challenging for readers unfamiliar with [24] (and not dive into the appendix proof). For example, it's hard to get much intuition why RingSample algorithm (Alg. 1) works unless a reader already knows the result of [24].




**Questions:**

Some other comments.
- I would recommend the authors to at least give a sketch of [24]'s idea before describing the RingSample algorithm.
- I recommend to have "coreset" formally defined somewhere.
- The notation $\mathrm{Cost}(X,Y)$ is never formally defined in the paper. Although I can guess what it is, it's better to define it somewhere as it is used frequently.
- line 224: Same as above for $\mathrm{Cost}_{|S|\leq k}(X, S)$.
- line 227-245: Many symbols defined here are not used at all in the main body. Please move them to the supplementary where they are actually used in the proof. Here they only distracts readers' attention.
- line 233-236: $R_I(C_i)$ and $R_O(C_i)$ have the same definition.
- line 243: I would write $G_{j, \mathrm{min}}$ and $G^O_b$ as union of some $R_{i,j}$'s rather than introducing a "$x$" there. It feels like you can take some $x$ from a $R_{i,j}$ while in fact it's all-or-nothing.
- line 248: Lemma 2.2: what is $G$?
- Section 3 (Experiment): My suggestion to the authors is to throw the experiment section to appendix or simply remove it, so you can have more space to enhance the readability of the theory part.

---

> ### Author Rebuttal · Authors · 2023-08-09
>
> > As mentioned in the Summary, the main idea seems to be an (arguably) straightforward adaptation of [24], which somewhat limits the novelty.
>
> Our main algorithm utilizes (1) a framework for sliding window algorithms using a new randomized online coreset construction along the lines of [8], (2) an online version of the coreset construction of [24], and (3) a consistent assignment across all times of the stream through an online facility location argument in the style of [30]. Although none of these steps are particularly complicated, developing the correct variations for each step and putting them together in a non-trivial manner is fairly novel/technical, e.g., the long line of previous work for clustering on sliding windows has missed this approach.
>
> > Also, I feel the experiment section is kinda redundant or even harmful to the paper's contribution. The proposed algorithm is not acutally implemented (in particular, the RingSample Alg, which is a core part of the proposed algorithm). It seems that the performance gain comes only from discarding points outside the sliding window, especially when compared with the histogram-based algorithm.
>
> Indeed, the main focus of our paper should be on the theoretical perspective of the algorithm and thus our experiments serve as a simple proof-of-concept illustrating the advantages of sampling-based algorithms over histogram-based algorithms. We will clarify the purpose of our experiments more clearly in the final version of our paper.
>
> > If the authors are able to implement their algorithm in a more meaningful way, and if it indeed produces better result on some realworld dataset, then I would be happy to increase my rating. Otherwise, I feel it's better to just remove the experiment section completely and present the paper as a theoretical contribution.
>
> We can move the experiments to the appendix in the full version so that the main body focuses on the theoretical contributions of the paper.
>
> > Lastly, I believe the writing can be improved a lot. It seems the draft may have been finalized hastily, with instances of noticeable cutting and pasting to meet the page limit. This renders the main body insufficiently self-contained and challenging for readers unfamiliar with [24] (and not dive into the appendix proof). For example, it's hard to get much intuition why RingSample algorithm (Alg. 1) works unless a reader already knows the result of [24].
>
> Thanks for the feedback. Much of the intuition and formalization for the coreset construction of [24], as well as our modifications, is currently in the appendix. If the paper is accepted, we will rework this intuition and incorporate it into the additional content page that is allowed for the main body of the paper.
>
> > line 248: Lemma 2.2: what is $G$?
>
> We will clarify $G$ to denote any fixed group, so that the first statement of Lemma 2.2 should read "Let $C$ be an $A$-approximate centroid set for any fixed group $G$"

---

### Author Rebuttal · Authors · 2023-08-09

We thank the reviewers for their thoughtful comments and valuable feedback. We appreciate the positive remarks, such as

- The problem studied is important and well-motivated. (Reviewer 8X17)
- It's nice to see that the result of [23] and [24] can be ported to the online setting. (Reviewer 8X17)
- The paper is able to identify an optimal choice of parameters, which results in an asymptotically optimal algorithm. (Reviewer 8X17)
- The paper is well-written in general, most parts are clear and well-organized. (Reviewer P4Gj)
- The results are interesting and significant...checking the key steps convince me the result should be correct. (Reviewer P4Gj)
- It is a plus for a theory paper to have experiments. The space consumption is reduced a lot with some compromise on the clustering performance. (Reviewer P4Gj)
- The presented result significantly improves the space complexity of previous work with $(1+\epsilon)$-approximation guarantee on the clustering quality in sliding window model. (Reviewer YLu7)
- In practical settings, since $\Delta$ is usually assumed to be bounded by $\text{poly}(n)$, the presented result can nearly match the lower bound space complexity required for any $(1+\epsilon)$-online coreset construction for $(k,z)$-clustering problem. (Reviewer YLu7)
- The extensive experimental results on real world datasets show that the proposed method is more efficient than previous ones. (Reviewer YLu7)
- The space $\frac{k}{\min(\varepsilon^4,\varepsilon^{2+z})}$ of $(1+\varepsilon)$-coreset for $(k,z)$-clustering in the sliding window almost matches the lower bound $\Omega\left(\frac{k}{\varepsilon^2}\log n\right)$. (Reviewer o3G5)
- Strict proof is provided and experimental results are sufficient. (Reviewer o3G5)
- The paper is well-written and well-organized. (Reviewer o3G5)

We provide our responses to the initial comments of each reviewer below. We hope our answers resolve any remaining questions, and we would be happy to participate in any conversations during the discussion phase.

---

### Decision · Program_Chairs · 2023-09-21

**Decision:**

Accept (poster)

**Comment:**

All reviewers appreciate the important problem of k-clustering in a natural setting: streaming with sliding windows. The paper gives a new algorithm with improved memory bound and fast runtime. Additionally the algorithm performs well in experiments.